# The function of juvenile–adult transition axis in female sexual receptivity of *Drosophila melanogaster*

Jing Li[1,2]*[†], Chao Ning[3,4†], Yaohua Liu[2,5†], Bowen Deng[6], Bingcai Wang[2,7], Kai Shi[2,7], Rencong Wang[2,7], Ruixin Fang[1], Chuan Zhou[1,2,7]

[1]Institute of Molecular Physiology, Shenzhen Bay Laboratory, Shenzhen, China; [2]State Key Laboratory of Integrated Management of Pest Insects and Rodents, Institute of Zoology, Chinese Academy of Sciences, Beijing, China; [3]National Laboratory of Biomacromolecules, New Cornerstone Science Laboratory, CAS Center for Excellence in Biomacromolecules, Institute of Biophysics, Chinese Academy of Sciences, Beijing, China; [4]CAS Key Laboratory of Genome Sciences and Information, Beijing Institute of Genomics, Chinese Academy of Sciences, Beijing, China; [5]Department of Plant Protection, Shanxi Agricultural University, Jinzhong, China; [6]Chinese Institute for Brain Research, Peking-Tsinghua Center for Life Sciences, Zhongguancun Life Sciences Park, Beijing, China; [7]University of Chinese Academy of Sciences, Beijing, China

*For correspondence:
lijing@szbl.ac.cn

[†]These authors contributed equally to this work

Competing interest: The authors declare that no competing interests exist.

**Abstract** Female sexual receptivity is essential for reproduction of a species. Neuropeptides play the main role in regulating female receptivity. However, whether neuropeptides regulate female sexual receptivity during the neurodevelopment is unknown. Here, we found the peptide hormone prothoracicotropic hormone (PTTH), which belongs to the insect PG (prothoracic gland) axis, negatively regulated virgin female receptivity through ecdysone during neurodevelopment in *Drosophila melanogaster*. We identified PTTH neurons as doublesex-positive neurons, they regulated virgin female receptivity before the metamorphosis during the third-instar larval stage. PTTH deletion resulted in the increased EcR-A expression in the whole newly formed prepupae. Furthermore, the ecdysone receptor EcR-A in pC1 neurons positively regulated virgin female receptivity during metamorphosis. The decreased EcR-A in pC1 neurons induced abnormal morphological development of pC1 neurons without changing neural activity. Among all subtypes of pC1 neurons, the function of EcR-A in pC1b neurons was necessary for virgin female copulation rate. These suggested that the changes of synaptic connections between pC1b and other neurons decreased female copulation rate. Moreover, female receptivity significantly decreased when the expression of PTTH receptor Torso was reduced in pC1 neurons. This suggested that PTTH not only regulates female receptivity through ecdysone but also through affecting female receptivity associated neurons directly. The PG axis has similar functional strategy as the hypothalamic–pituitary–gonadal axis in mammals to trigger the juvenile–adult transition. Our work suggests a general mechanism underlying which the neurodevelopment during maturation regulates female sexual receptivity.

## eLife assessment

The aim of this **valuable** study is to uncover developmental roles of the neuropeptide prothoracicotropic hormone (PTTH) and ecdysone, which later regulate female receptivity of *Drosophila melanogaster*. The work combines spatially and temporally restricted genetic manipulation with behavior quantification to explore these molecular pathways and the neuronal substrates participating in the

control of female sexual receptivity. At present, the implication of both signaling pathways in this process is **convincing** but the strength of the evidence is **incomplete** to support the main claim that PTTH pathway controls female sexual receptivity through the function of ecdysone in pC1 neurons.

## Introduction

The success of copulation is important for the reproduction of a species. *Drosophila melanogaster* provides a powerful system to investigate the neuronal and molecular mechanism of sexual behaviors. Females decide to mate or not according to their physiological status and the environmental condition (*Dickson, 2008*). Sexually mature adult virgin females validate males after sensing the courtship song and male-specific sex pheromone, receive courtship with pausing and opening the vaginal plate (VPO) (*Ferveur, 2010*; *Greenspan and Ferveur, 2000*; *Hall, 1994*; *Wang et al., 2021*). If the female is not willing to mate, she may kick her legs, flick her wings, or extrude the ovipositor to deter males (*Connolly and Cook, 1973*). Mated females reject males for several days after mating mainly through more ovipositor extrusion (OE) and less VPO (*Fuyama and Ueyama, 1997*; *Wang et al., 2021*). These options need the establishment of neural circuits for female sexual receptivity. However, the associated mechanism of neural maturation and the effect of neural maturation on female sexual receptivity are little known.

*doublesex* (*dsx*) and *fruitless* (*fru*) are the terminal genes in sex determination regulatory hierarchy. They specify nearly all aspects of somatic sexual differentiation, including the preparation for sexual behaviors (*Dickson, 2008*; *Manoli et al., 2013*; *Manoli et al., 2006*; *Mellert et al., 2012*; *Pavlou and Goodwin, 2013*; *Siwicki and Kravitz, 2009*; *Yamamoto, 2007*; *Yamamoto and Koganezawa, 2013*). In males, expression of male-specific Fru$^M$ (*Billeter et al., 2006*; *Demir and Dickson, 2005*; *Hall, 1978*; *Manoli et al., 2005*; *Stockinger et al., 2005*) and male-specific Dsx$^M$ (*Kohatsu et al., 2011*; *Pan and Baker, 2014*; *Pan et al., 2011*; *Rideout et al., 2010*) is important for male courtship behaviors. In females, although functional Fru protein is not translated, neurons with Dsx$^F$ or *fru* P1 promoter regulate some aspects of the female sexual behaviors (*Kvitsiani and Dickson, 2006*; *Rideout et al., 2010*). *Fru and dsx* are involved in regulating the sexual dimorphism during neuro-development (*Yamamoto and Koganezawa, 2013*). For instance, the sexual dimorphism of P1 and mAL neurons which are all associated with male courtship and aggression behaviors (*Clowney et al., 2015*; *Hoopfer et al., 2015*; *Kimura et al., 2008*; *Kohatsu et al., 2011*; *Pan et al., 2012*; *Sengupta et al., 2022*; *von Philipsborn et al., 2011*) is the result of regulation by Dsx and/or Fru (*Ito et al., 2012*; *Kimura et al., 2008*). In the cis-vaccenyl acetate (cVA) pathway, which induces the courtship inhibiting in males (*Kurtovic et al., 2007*; *Wang and Anderson, 2010*), the first-order to the fourth-order components are all fru-Gal4-positive neurons and are either male-specific or sexually dimorphic (*Ruta et al., 2010*). However, the role of Dsx$^F$ in neurodevelopment associated with female sexual behaviors is little understood.

During postembryonic development, the PG axis triggers the juvenile–adult transition, similar to the function of hypothalamic–pituitary–gonadal (HPG) axis in mammals (*Herbison, 2016*; *Pan and O'Connor, 2019*). Hormones of the PG axis act to transform the larval nervous system into an adult version (*Truman and Riddiford, 2023*). Ecdysone belonging to the PG axis is the prime mover of insect molting and metamorphosis and is involved in all phases of neurodevelopment, including neurogenesis, pruning, arbor outgrowth, and cell death (*Truman and Riddiford, 2023*). The neurons read the ecdysteroid titer through two isoforms of the ecdysone receptor, EcR-A and EcR-B1, according to spatial and temporal conditions in the central nervous system (CNS) (*Riddiford et al., 2000*; *Truman et al., 1994*). EcR-A is required in fru P1-expressing neurons for the establishment of male-specific neuronal architecture, and ecdysone receptor deficient males display increased male–male courtship behavior (*Dalton et al., 2009*; *Ganter et al., 2007*). However, how ecdysone regulates the neurodevelopment associated with female sexual receptivity, especially the fru$^+$ and dsx$^+$ neurons, is unknown.

Much of studies to understand female sexual receptivity has focused on its regulation. How a female respond to males is highly dependent on whether or not she has previously mated. In virgin females, dsx$^+$ pCd neurons respond to the cVA, while dsx$^+$ pC1 neurons also respond to male courtship song (*Zhou et al., 2014*). The receptive females open the vaginal plate (VPO) through activation of the dsx$^+$ vpoDN neurons (*Wang et al., 2021*). After mated, sex peptide in the seminal fluid binds to the fru$^+$ dsx$^+$ sex peptide sensory neurons in the female uterus. Then neuronal activity in the dsx$^+$

sex peptide abdominal ganglion neurons of the ventral nerve cord and in the pC1 neurons is reduced (*Avila et al., 2011*; *Feng et al., 2014*; *Häsemeyer et al., 2009*; *Kubli, 2003*; *Wang et al., 2020b*; *Yang et al., 2009*; *Zhou et al., 2014*). Therefore, the sexual receptivity is reduced with less VPO and more OE which is controlled by dsx+ DpN13 neurons (*Wang et al., 2020a*). In addition, neuropeptides and monoamines play a critical role in regulation of the female receptivity. The neuropeptides Drosulfakinin, myoinhibitory peptides and SIFamide are involved in female sexual receptivity (*Jang et al., 2017*; *Terhzaz et al., 2007*; *Wang et al., 2022*). As monoamines, dopamine, serotonin, and octopamine are pivotal to female sexual behaviors (*Ishimoto and Kamikouchi, 2020*; *Ma et al., 2022*; *Neckameyer, 1998*; *Rezával et al., 2014*). So far, the identified neuropeptides and monoamines modulating female sexual receptivity all function during the adult stage. However, whether neuropeptides or monoamines regulate the establishment of neural circuits for female sexual receptivity is unknown.

To explore the factors that regulate *Drosophila* virgin female receptivity especially during neurodevelopment, we did a knock-out screen including most of chemoconnectome (CCT) members. We discovered a requirement for the prothoracicotropic hormone (PTTH) during postembryonic development for virgin female receptivity. We also found that PTTH neurons expressing PTTH are dsx+ neurons. PTTH, a brain-derived neuropeptide hormone, is the primary promoter of the synthesis of steroid hormone 20-hydroxyecdysone (20E) (*McBrayer et al., 2007*; *Rewitz et al., 2009*). Indeed, the enhanced virgin female receptivity due to the loss of PTTH could be rescued through feeding 20E to the third-instar larvae. Because 20E acts through its receptor EcR (*Riddiford et al., 2000*), we then tested the function of EcR in pC1 neurons which encode the mating status of females (*Zhou et al., 2014*). The reduced EcR-A expression in pC1 neurons resulted in the abnormal anatomical pattern of pC1 neurons and the reduced female copulation rate. This may be explained by the increased EcR-A in newly formed prepupae resulted from the PTTH deletion. Furthermore, the decreased female copulation rate was due to the reduced EcR-A in pC1b neurons. Besides, we detected the inhibited female receptivity when PTTH receptor torso was decreased in pC1 neurons, suggested the direct function of PTTH on other dsx+ neurons to regulate female receptivity. Thus, in addition to demonstrating the function of PTTH in virgin female receptivity during neurodevelopment, our study identified the necessary role of the normal pC1b neural morphology in virgin female receptivity.

## Results
### PTTH modulates virgin female receptivity

In *Drosophila*, neuropeptides and monoamines, belonging to the CCT (the entire set of neurotransmitters, neuromodulators, neuropeptides, and their receptors underlying chemotransmission) (*Deng et al., 2019*), play a critical role in regulation of the female receptivity. To explore the factors that regulate virgin female receptivity especially during neurodevelopment, we screened 108 CCT knock-out lines generated by the CRISPR–Cas9 system (*Deng et al., 2019*) (unpublished data). The result showed that PTTH might regulate virgin female receptivity. The deletion mutant *Ptth^Delete* removed part of the 5′ UTR and almost all coding sequence and is a protein null (*Figure 1A*). We confirmed the PTTH knock-out flies by using PCR (Polymerase Chain Reaction) analysis at the PTTH locus in genomic DNA samples (*Figure 1B*), by using RT-PCR (Real-time PCR) to identify the loss of PTTH transcripts in cDNA samples (*Figure 1C*) and by detecting the immunoreactivity of PTTH in the central brain (*Figure 1—figure supplement 1A*). Primers used are listed in *Supplementary file 1*. PTTH immunoreactivity was found in the brain of wild-type and heterozygous flies (*Figure 1—figure supplement 1A1, 1A3*), but was absent in homozygous *Ptth^Delete* flies (*Figure 1—figure supplement 1A2*). As the previous study, the *Ptth^Delete* larvae lacking PTTH undergo metamorphosis with about 1 day delay compared with the wild-type control (*Shimell et al., 2018*) (data not shown). Besides, the *Ptth^Delete* adult male and female flies had the significant increased weight than wild-type flies (*Figure 1—figure supplement 1B*). This is also consistent with that PTTH regulates developmental timing and body size in *Drosophila* (*McBrayer et al., 2007*; *Shimell et al., 2018*).

To confirm the function of PTTH, we tested virgin female receptivity of *Ptth^Delete* female flies. We found that the virgin female losing PTTH had significantly higher copulation rate and shorter latency to copulation than wild-type flies (*Figure 1D–G*). In addition, the *Ptth^Delete* flies had higher copulation rate and lower latency to copulation compared to heterozygous null mutant females within 2 days

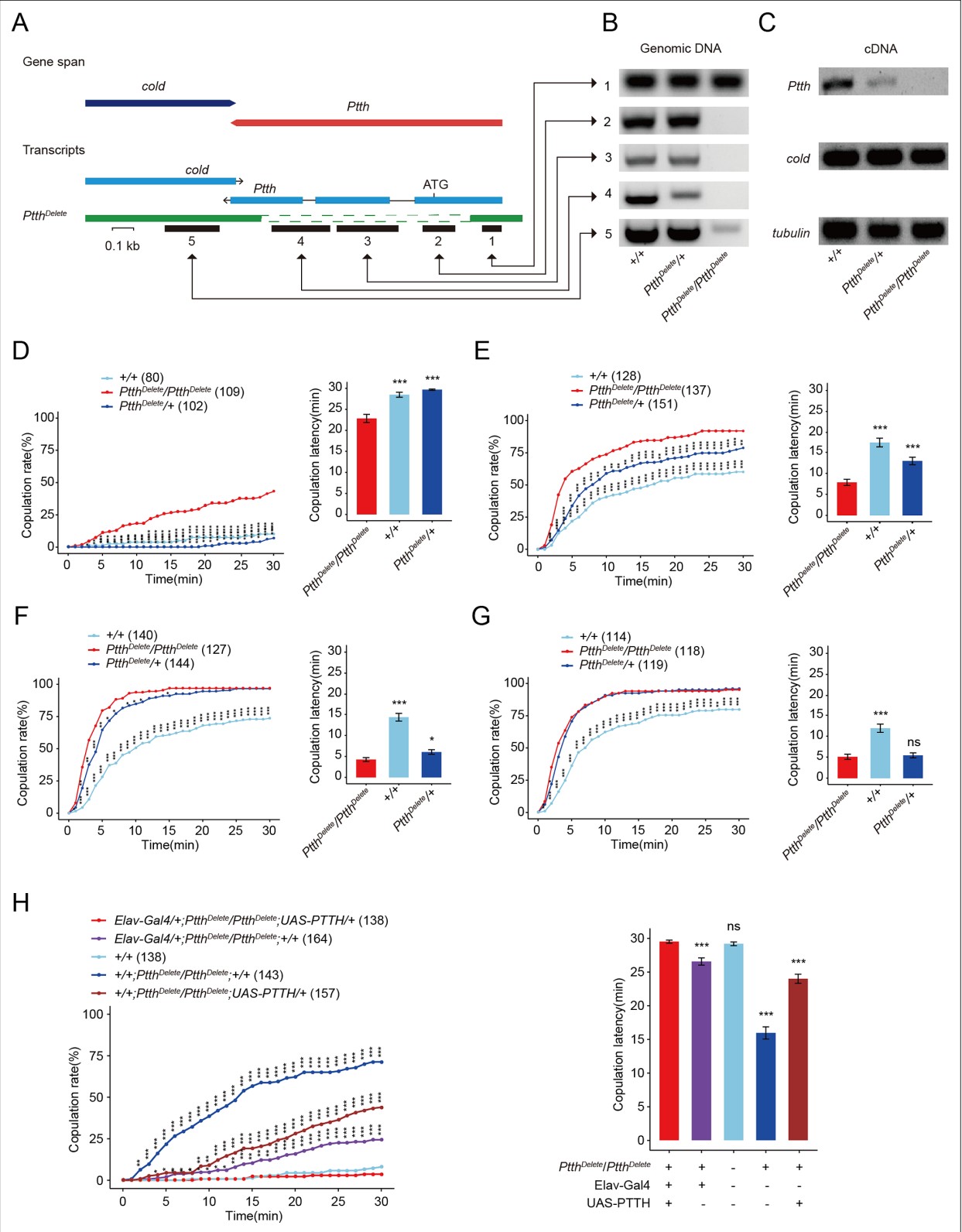

**Figure 1.** *Ptth* null mutants have increased virgin female receptivity. (**A–C**) Generation and validation of a 974-bp deletion mutant of the *Ptth* gene. The 5' UTR and almost all coding sequence were deleted. The deletion was confirmed through PCR analysis at the prothoracicotropic hormone (PTTH) locus in genomic DNA samples (**B**), and through RT-PCR to identify the loss of PTTH transcripts in cDNA samples of wandering larvae (**C**). Virgin female receptivity of *Ptth* null mutants on the first (**D**), second (**E**), third (**F**), and sixth day (**G**), respectively. The comparison referred to *Ptth*$^{Delete}$/*Ptth*$^{Delete}$.

*Figure 1 continued on next page*

*Figure 1 continued*

(**H**) Enhanced virgin female receptivity of *ΔPtth* null mutants was rescued by elav-Gal4 driving UAS-PTTH. The increased copulation rate and decreased latency to copulation on the first day after eclosion were rescued to the comparable level of control. The comparison referred to *elav-Gal4/+; Ptth^Delete^/Ptth^Delete^;UAS-PTTH/+*. The copulation latency and copulation rate of *elav-Gal4/+; Ptth^Delete^/Ptth^Delete^* are higher and lower than *Ptth^Delete^/Ptth^Delete^;UAS-PTTH/+*, respectively. The number of female flies paired with wild-type males is displayed in parentheses. For the copulation rate, chi-square test is applied. For the latency to copulation, Kruskal–Wallis ANOVA (Analysis of Variance) and post hoc Mann–Whitney $U$ tests are applied. Error bars indicate SEM (Mean standard error). *$p < 0.05$, ***$p < 0.001$, ns indicates no significant difference.

The online version of this article includes the following source data and figure supplement(s) for figure 1:

**Source data 1.** Photo of nucleic acid electrophoresis and copulation time.

**Figure supplement 1.** Prothoracicotropic hormone (PTTH) expression, weight, attractiveness, and locomotion behavior of *Ptth* null mutant virgin females.

**Figure supplement 1—source data 1.** Body weight, courtship index, and walking speed.

**Figure supplement 2.** Effect of the expression of prothoracicotropic hormone (PTTH) on female receptivity.

**Figure supplement 2—source data 1.** Copulation time.

(*Figure 1D, E*) and within 3 days, respectively (*Figure 1D–F*). The enhanced virgin female receptivity had no relationship either with the attractivity or with the locomotion activity of virgin females (*Figure 1—figure supplement 1C–E*). These results suggested that PTTH deletion regulates virgin female receptivity in a dose-dependent manner. Female receptivity increases with the increase of age after eclosion, not only for wild-type flies but also PTTH mutants. At the first day after eclosion (*Figure 1D*), maybe the loss of PTTH in *PTTH^Delete^/+* flies is not enough for sexual precocity as *PTTH^Delete^/PTTH^Delete^*. At the second day after eclosion and after (*Figure 1E–G*), the loss of PTTH in *PTTH^Delete^/+* flies is enough for sexual precocity compared with wild-type flies. However, After the second day of adult, female receptivity of all genotype flies increases sharply. At the third day of adult and after, female receptivity of *PTTH^Delete^/PTTH^Delete^* reaches the peak and the receptivity of *PTTH^Delete^/+* reaches more nearly to *PTTH^Delete^/PTTH^Delete^* when flies get older (*Figure 1F, G*). However, the overexpression through PTTH-Gal4>UAS-PTTH is also not sufficient to change female receptivity (*Figure 1—figure supplement 2A*). Similarly, decreased expression of PTTH through PTTH-Gal4>UAS-PTTH-RNAi or dsx-Gal4>UAS-PTTH-RNAi did not result in the similar phenotype to that of *PTTH^Delete^/PTTH^Delete^* (*Figure 1—figure supplement 2B, C*). It is possible that both decreasing and increasing PTTH expression are not sufficient to change female receptivity.

Furthermore, we carried out genetic rescue experiments to further confirm the function of PTTH in modulating virgin female receptivity. We used the pan-neuronal driver elav-Gal4 to drive UAS-PTTH expression in PTTH mutant background, although elav-Gal4 did not express in PTTH neurons (*Figure 1—figure supplement 2D*). We detected the PTTH signals using PTTH antibody in the rescued female brains (*Figure 1—figure supplement 1A4*). We found that neuron-specific expression of PTTH could restore the enhanced copulation rate and shorter latency to copulation in *PTTH^Delete^/PTTH^Delete^* virgin females (*Figure 1H*). Except for the projection of axons to PG gland, PTTH also carries endocrine function to regulate light avoidance of larvae (*Yamanaka et al., 2013*). The overexpressed PTTH in other neurons through elav-Gal4>UAS-PTTH may act on the PG gland through endocrine function and then induce the ecdysone synthesis and release. In summary, these results suggested that PTTH regulates virgin female receptivity.

## Dsx⁺ PTTH neurons regulate virgin female receptivity

We used new Ptth-Gal4 and Ptth-LexA which inserts Gal4 or LexA sequence before the stop codon of the *Ptth* gene (*Deng et al., 2019*) to label and manipulate PTTH neurons expressing PTTH. The labeled neurons were the same as reported before (*McBrayer et al., 2007*; *Yamanaka et al., 2013*), a pair of bilateral neurosecretory cells in the brain directly innervating the prothoracic gland during the larval stage (*Figure 2A* and *Figure 2—figure supplement 1A*). The newly emerged flies had the similar anatomical pattern with that of the larval stage (*Figure 2B* and *Figure 2—figure supplement 1B*). However, while the prothoracic gland cells are gradually degenerating during pharate adult development (*Dai and Gilbert, 1991*; *Roy et al., 2018*), the pattern of PTTH neurons labeled by Ptth-Gal4>UAS-mCD8GFP gradually could not be found after the 10th hour after eclosion (*Figure 2—figure supplement 2*).

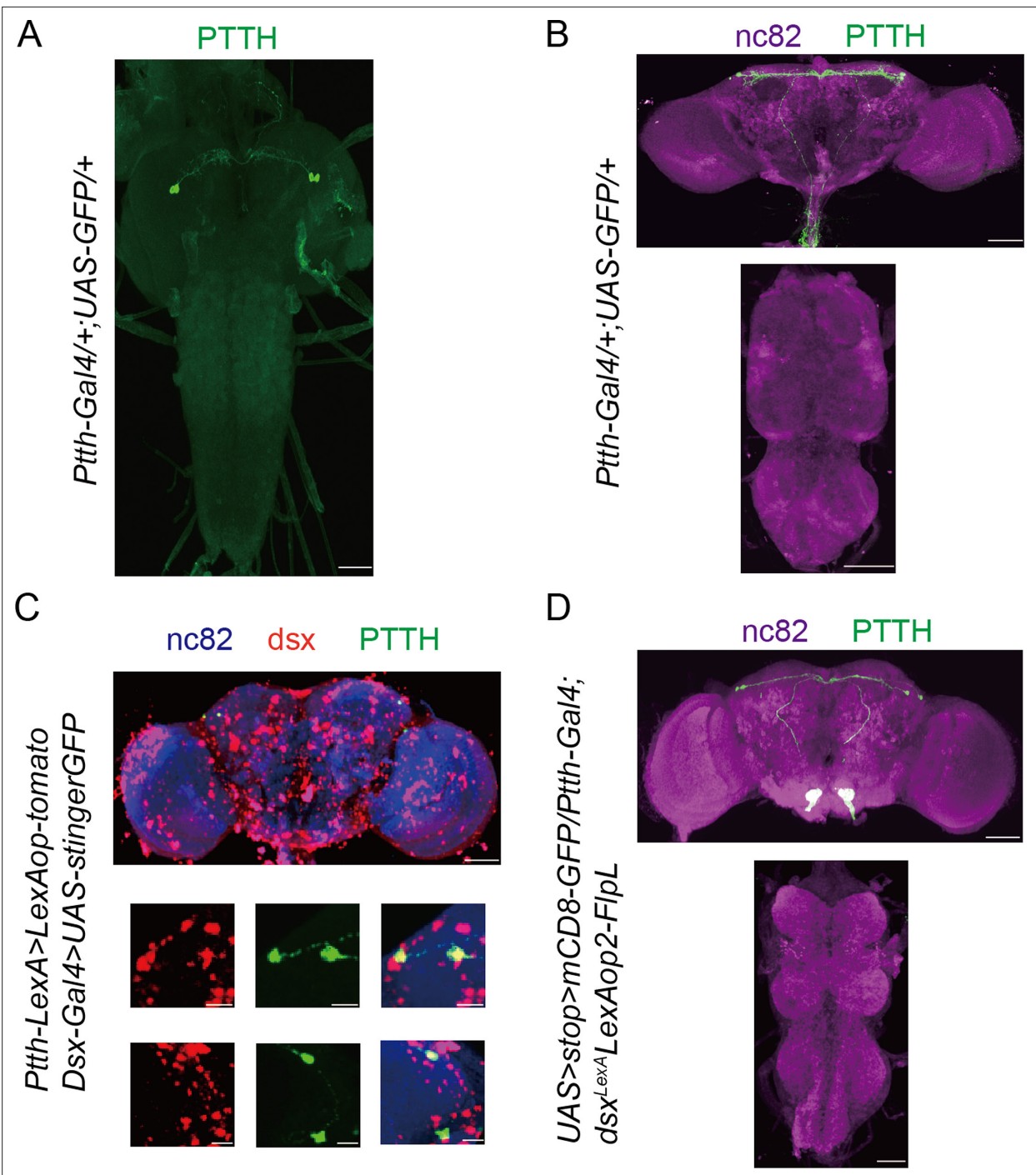

**Figure 2.** Prothoracicotropic hormone (PTTH) neurons are doublesex-positive neurons. Expression pattern of Ptth-Gal4 revealed by anti-GFP in larvae central nervous system (CNS) (**A**) and adult brain (**B**). Representative of five female flies. Scale bars, 50 µm. (**C**) All PTTH neurons were colabeled by dsx-Gal4 driving UAS-GFP-Stinger (red) and Ptth-LexA driving LexAop-tomato (green). Representative of five female brains. Scale bars, 50 and 5 µm (zoom-in). (**D**) All PTTH neurons were *Ptth* and *Dsx* co-expressing, labeled by intersectional strategy. The larvae flies were the wandering ones. The adult flies were within 10-hr-old adults. Representative of five female brains. Scale bars, 50 µm.

The online version of this article includes the following source data and figure supplement(s) for figure 2:

**Figure supplement 1.** The function of prothoracicotropic hormone (PTTH) neurons in female receptivity.

**Figure supplement 1—source data 1.** Copulation time.

**Figure supplement 2.** The anatomical pattern of prothoracicotropic hormone (PTTH) neurons expressing PTTH at different developmental stages.

*Figure 2 continued on next page*

*Figure 2 continued*

**Figure supplement 3.** Prothoracicotropic hormone (PTTH) neurons expressing PTTH do not regulate virgin female copulation rate during adult stage.

**Figure supplement 3—source data 1.** Copulation time.

Most identified neurons associated with female sexual behaviors express *doublesex* gene. We asked whether PTTH neurons are a part of the doublesex circuitry or not. Double labeling of dsx-LexA and Ptth-Gal4 neurons (LexAop-tomato,UAS-stinger-GFP/Ptth-LexA;dsx-Gal4/+) revealed that PTTH neurons are all doublesex-positive (*Figure 2C*). We then used an intersectional strategy to visualize overlapped expression between dsx-LexA and Ptth-Gal4 (*UAS > stop > myrGFP/+;LexAop2-FlpL,dsx-LexA/Ptth-Gal4*). We observed all PTTH neurons with GFP signals (*Figure 2D*). These results suggested that PTTH neurons are dsx⁺ neurons. Furthermore, we wanted to know whether Dsx$^F$ regulates female receptivity in PTTH neurons. We decreased the Dsx$^F$ expression in PTTH neurons and did not detect significantly changed female receptivity (*Figure 2—figure supplement 1C*). We supposed that PTTH neurons have some relationship with other Dsx$^F$-positive neurons which regulate female receptivity.

We then analyzed whether PTTH neurons are involved in the modulation of virgin female receptivity. First, we activated PTTH neurons transiently in adult virgin females by driving the temperature-sensitive activator dTrpA1 (*Hamada et al., 2008*) using Ptth-Gal4. PTTH neurons were activated at 29°C compared with the control treatment at 23°C. No significantly different copulation rate or latency to copulation was detected (*Figure 2—figure supplement 3A–C*). This suggested that PTTH neurons do not regulate virgin female receptivity during the adult stage.

To identify the detail time for the function of PTTH neurons in virgin female receptivity, we inactivated PTTH neurons through kir2.1 under the control of the temporal and regional gene expression targeting system (*McGuire et al., 2004*). When the experiment was done at the larval stage is the only situation when the controls were both different from the experimental (*Figure 2—figure supplement 1D*). However, when PTTH neurons were inactivated during whole pupal or whole adult stages, virgin female copulation rate did not change significantly (*Figure 2—figure supplement 1D*). Furthermore, we activated PTTH neurons at different stages overlapping the postembryonic larval developmental time using dTrpA1 (*Figure 3A*). Stage 1 was from the first-instar larvae to 6 hr before the third-instar larvae. Stage 2 was from 6 hr before the third-instar larvae to the end of the wandering larvae (the start of prepupa stage). Stage 3 was from the start of prepupa stage to the end of the second day of the pupal stage. Stage 4 was from the end of the second day of the pupal stage to the eclosion of adults. The copulation rate did not change significantly when activating PTTH neurons during the stage 1, 3, or 4 (*Figure 3B, D, E*). However, we found the significant lower copulation rate and the longer latency to copulation only when PTTH neurons were activated during the stage 2 (*Figure 3C, F*). The defected copulation was not due to a lower locomotion activity of virgin females (*Figure 3G*). Taken together, our findings indicated that the activity of dsx⁺ PTTH neurons negatively regulate virgin female receptivity during the stage from the start of the third instar to the end of wandering stage.

## PTTH modulates virgin female receptivity through ecdysone

The third-intar larval stage is the critical stage for the initiation of metamorphosis involving the synthesis of ecdysone (*Imura et al., 2020*; *Lavrynenko et al., 2015*; *Shimell et al., 2018*). To test whether PTTH regulates virgin female receptivity through regulating the synthesis of ecdysone, we rescued the enhanced female receptivity by feeding 20E to the third-intar larval *Ptth$^{Delete}$* flies. The enhanced copulation rate and shorter latency to copulation of the *Ptth$^{Delete}$* flies were rescued to the comparable level of wild-type females (*Figure 4*). Furthermore, the wild-type females fed by 20E had no significantly different copulation rate and latency to copulation compared with the wild-type females fed by the same volume of 95% ethanol which is the solvent of 20E (*Figure 4*). This suggested that PTTH regulates virgin female receptivity through the titer of ecdysone.

## Ecdysone receptor EcR-A in pC1 neurons regulates virgin female copulation rate

Given that PTTH regulates virgin female receptivity through ecdysone which acts on its receptor EcR, we then asked whether ecdysone regulates the function of neurons associated with virgin female receptivity through EcR. pC1 and vpoDN neurons are two main dsx⁺ neurons involved in virgin female

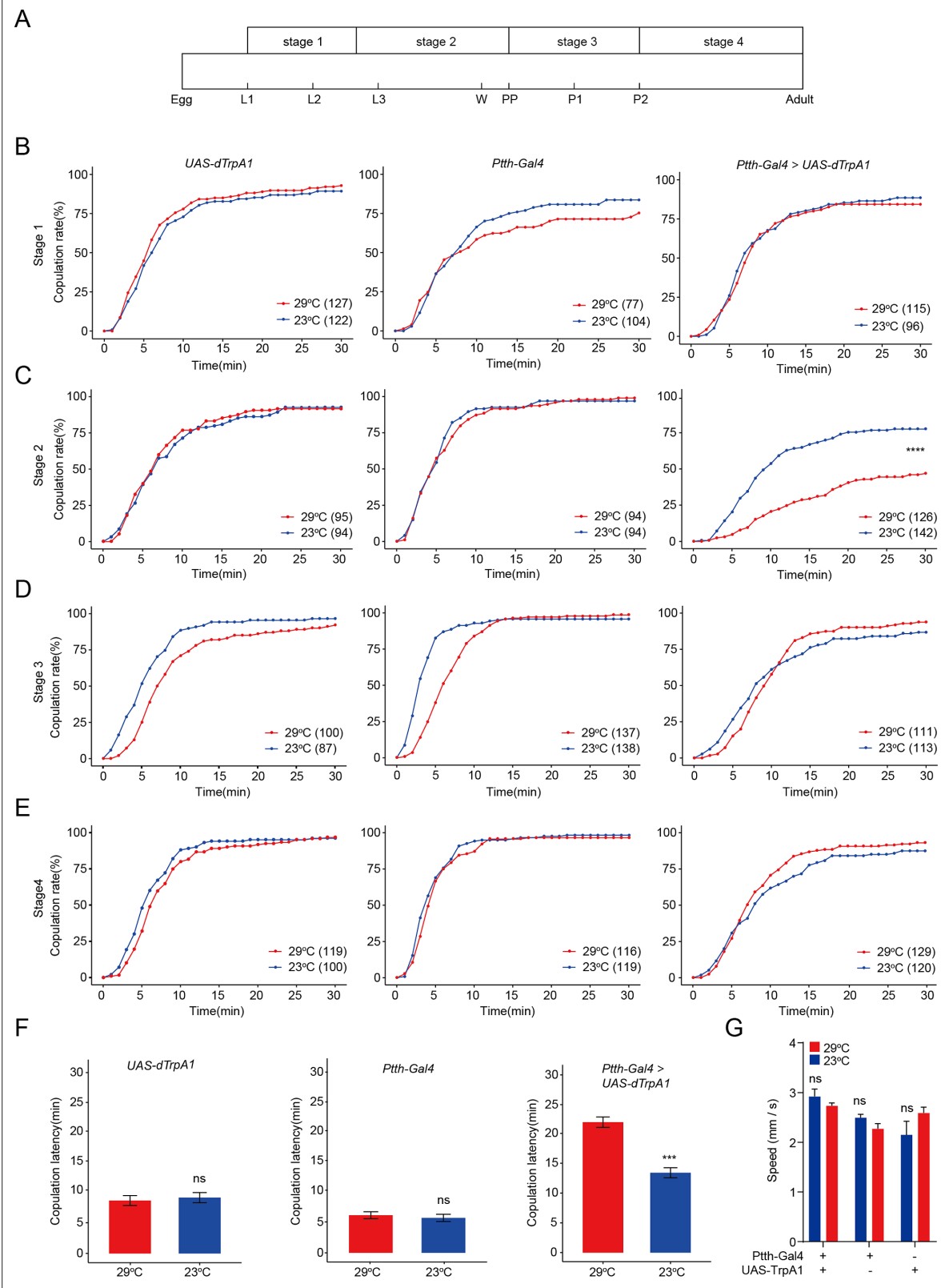

**Figure 3.** Activation of prothoracicotropic hormone (PTTH) neurons expressing PTTH during the third-instar larvae inhibits virgin female receptivity. (**A**) Four developmental stages of *Drosophila* before eclosion when PTTH neurons were thermogenetic activated by dTrpA1. L1, L2, and L3: start of three larval stages, W: start of wandering stage, Pp: puparium formation, P1 and P2: start of the first and second day of pupal stage. (**B–E**) Ptth-Gal4 driving UAS-dTrpA1 activated PTTH neurons at 29°C. Activation of PTTH neurons at the stage 2 significantly decreased copulation rate (**C**), but not at the stage

*Figure 3 continued on next page*

*Figure 3 continued*

1 (**B**), stage 3 (**D**), and stage 4 (**E**). (**F**) Activation of PTTH neurons at the stage 2 significantly increased the latency to copulation. (**G**) Mean velocity had no significant change when PTTH neurons were activated during the stage 2 compared with control females (ns = not significant, Kruskal–Wallis ANOVA and post hoc Mann–Whitney *U* tests, mean ± SEM, *n* = 8–12). The comparison referred to *Ptth-Gal4/UAS-dTrpA1*. Female flies were 4-day-old adults. The number of female flies paired with wild-type males is displayed in parentheses. For the copulation rate, chi-square test is applied. For the latency to copulation, Mann–Whitney *U* test is applied. Error bars indicate SEM . ***p < 0.001, ****p < 0.0001, ns indicates no significant difference.

The online version of this article includes the following source data for figure 3:

**Source data 1.** Copulation time and walking speed.

receptivity (*Wang et al., 2021*; *Zhou et al., 2014*). pC1 neurons encode the mating status of female flies, vpoDN neurons regulate the VPO when females attend to accept males. EcR-A and EcR-B1 are the two prominently expressed ecdysone receptors in the CNS (*Riddiford et al., 2000*). First, we tested the expression of EcR-A and EcR-B1 in these two neurons on the second day of the pupal stage when ecdysone functions as the main mover in the metamorphosis (*Dalton et al., 2009*; *Truman et al., 1994*). The GFP signals labeled by pC1-ss2-Gal4 and vpoDN-ss1-Gal4 were merged well with the signals of both EcR-A and EcR-B1 antibodies, respectively (*Figure 5—figure supplement 1*). This revealed that EcR-A and EcR-B1 express in both pC1 and vpoDN neurons. We then tested the function of EcR in pC1 and vpoDN neurons through reducing the expression of EcR-A and EcR-B1, respectively. We used the split-Gal4 for pC1 and vpoDN neurons to drive the UAS-EcR-RNAi. First, we reduced the expression of all EcR isoforms in pC1 neurons, this decreased the copulation rate and prolonged the latency to copulation significantly (*Figure 5—figure supplement 2A*). Furthermore, we reduced the expression of EcR-A in pC1 neurons. The virgin female had the significant lower copulation rate and longer latency to copulation (*Figure 5A*). The reduced copulation rate had no relationship with the attractivity (*Figure 5E*) and the locomotion activity of virgin females (*Figure 5—figure supplement*

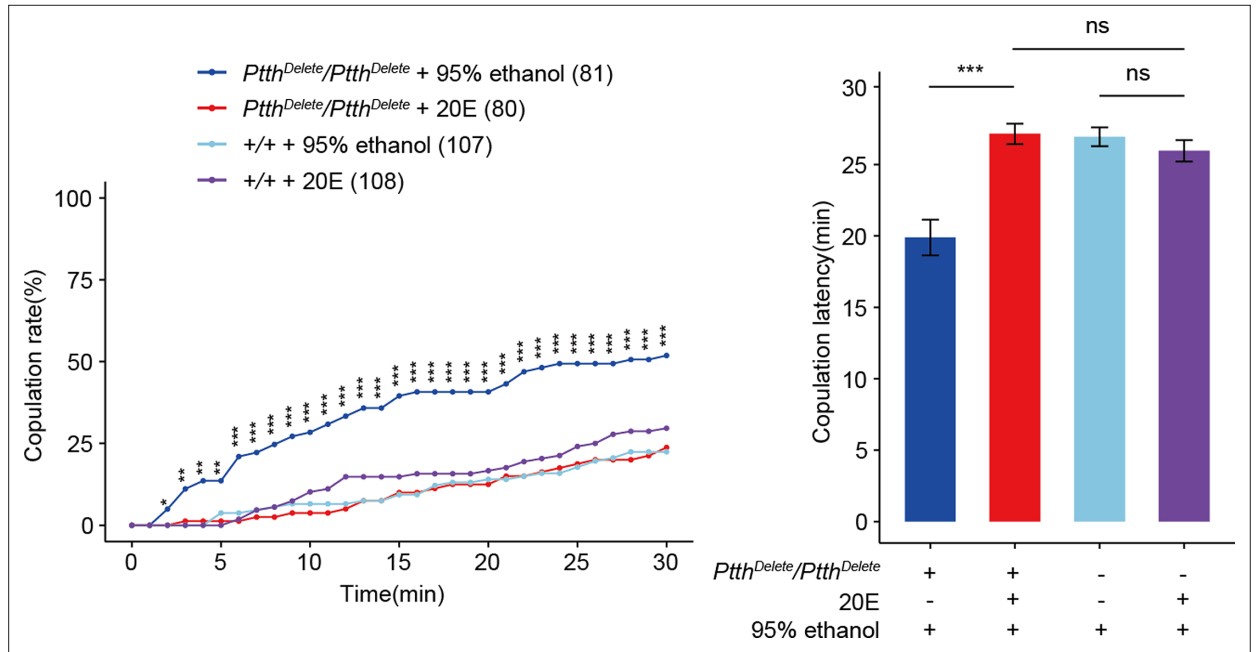

**Figure 4.** Feeding 20E restores virgin female receptivity of *Ptth* null mutant flies. The increased copulation rate and decreased latency to copulation of the 24-hr-old *ΔPtth* flies were rescued to the comparable level of wild-type females by feeding 20E to the third-instar larval *ΔPtth* flies. The wild-type larval females fed by 20E had no significantly different copulation rate and latency to copulation compared with the wild-type females fed by the same volume of 95% ethanol which is the solvent of 20E. The comparison referred to *Ptth^Delete/PtthD^elete* + 20E. Female flies were 1-day-old. The number of female flies paired with wild-type males is displayed in parentheses. For the copulation rate, chi-square test is applied. For the latency to copulation, Kruskal–Wallis ANOVA and post hoc Mann–Whitney *U* tests are applied. Error bars indicate SEM. *p < 0.05, **p < 0.01, ***p < 0.001, ns indicates no significant difference.

The online version of this article includes the following source data for figure 4:

**Source data 1.** Copulation time.

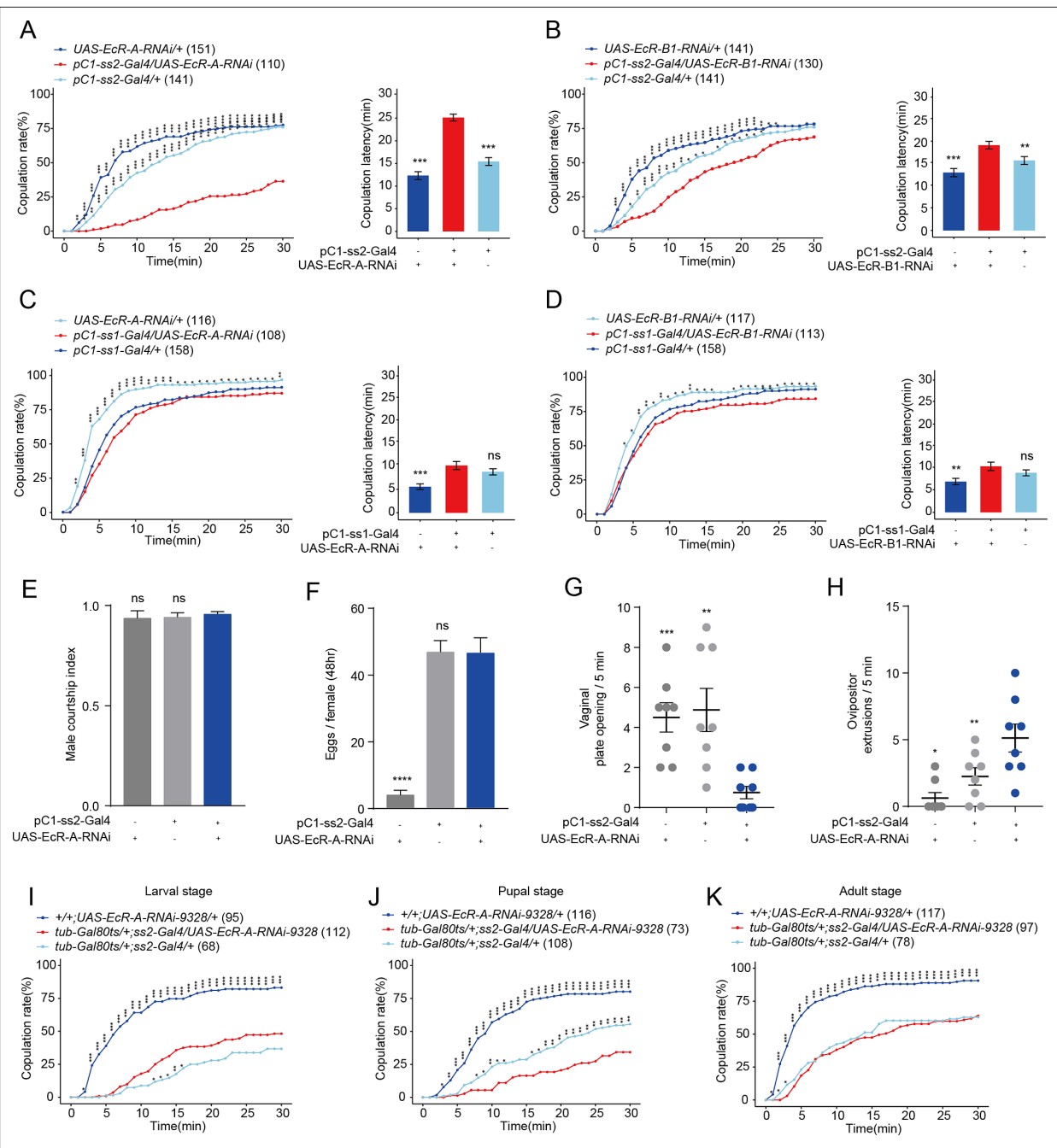

**Figure 5.** Virgin females with reduced EcR-A in pC1 neurons have reduced sexual receptivity. (**A**) Knock-down of EcR-A in pC1 neurons driven by pC1-ss2-Gal4 significantly decreased the copulation rate and increased the latency to copulation. (**B**) Knock-down of EcR-B1 in pC1 neurons driven by pC1-ss2-Gal4 significantly prolonged the latency to copulation. Knock-down of EcR-A (**C**) or EcR-B1 (**D**) in pC1 neurons driven by pC1-ss1-Gal4 did not affect the copulation rate or the latency to copulation. (**E**) Courtship index of wild-type males toward a female with the indicated genotype (*n* = 8). (**F**) The number of eggs laid by virgin females during the third to fourth day after eclosion when EcR-A was knocked down in pC1 neurons (*n* = 17–36). The UAS-EcR-A-RNAi control causes a massive decrease in female fertility. (**G**) Knock-down of EcR-A in pC1 neurons decreased the opening of vaginal plate of virgin females compared with controls (*n* = 8). (**H**) Knock-down of EcR-A in pC1 neurons increased the ovipositor extrusion of virgin females compared with controls (*n* = 8). (**I–K**) Virgin female copulation rate when EcR-A was knocked down in pC1 neurons temporally restricted by shifts from 18°C to 30°C. EcR-A was knocked down during the whole larval (**I**), pupal (**J**), and adult (**K**) stages, respectively. When the experiment was done at the pupal stage is the only situation when the controls were both different from the experimental (**J**). The comparison referred to flies with decreased EcR isoform in pC1 neurons. Female flies for behavioral assay were 4- to 6-day-old adults. The number of female flies paired with wild-type males is displayed in parentheses. For the copulation rate, chi-square test is applied. For other comparisons, Kruskal–Wallis ANOVA and post hoc Mann–Whitney *U* tests are applied. Error bars indicate SEM. *$p < 0.05$, **$p < 0.01$, ***$p < 0.001$, ****$p < 0.0001$, ns indicates no significant difference.

*Figure 5 continued on next page*

*Figure 5 continued*

The online version of this article includes the following source data and figure supplement(s) for figure 5:

**Source data 1.** Copulation time, courtship index, number of eggs, number of vaginal plate opening (VPO), and number of ovipositor extrusion (OE).

**Figure supplement 1.** Expression of EcR-A and EcR-B1 in pC1 and vpoDN neurons.

**Figure supplement 2.** Reduced EcR in pC1 neurons reduces virgin female receptivity.

**Figure supplement 2—source data 1.** Copulation time and walking speed.

**Figure supplement 3.** Reduced EcR in vpoDN neurons has no effect on virgin female receptivity.

**Figure supplement 3—source data 1.** Copulation time.

**Figure supplement 4.** Reduced EcR-A in pC1d neurons has no effect on virgin female receptivity.

**Figure supplement 4—source data 1.** Copulation time.

*2B*). When reducing the expression of EcR-B1 in pC1 neurons, virgin females had the significant longer latency to copulation but the comparable copulation rate to controls (*Figure 5B*). However, reducing the expression of EcR-A (*Figure 5—figure supplement 3A–C*) and EcR-B1 (*Figure 5—figure supplement 3D–F*) using three split vpoDN-Gal4s in vpoDN neurons all did not affect virgin female receptivity. This suggested that the expression of EcR-A in pC1 neurons regulates virgin female copulation rate, but EcR isoforms in vpoDN neurons do not modulate virgin female receptivity.

Two split-Gal4 drivers for pC1 neurons had been obtained previously. pC1-ss1-Gal4 labels pC1-a, -c, and -e neurons, and pC1-ss2-Gal4 labels all pC1-a, -b, -c, -d, and -e neurons (*Wang et al., 2020b*). We also tested virgin female receptivity when EcR-A or EcR-B1 were reduced in pC1-a, -c, and -e neurons simultaneously using pC1-ss1-Gal4, respectively. While the copulation rate or the latency to copulation did not change significantly (*Figure 5C, D*). This suggested that, pC1b only, or both pC1b and pC1d neurons is necessary for the functions of EcR-A and EcR-B1 in pC1 neurons on virgin female receptivity. Whether pC1d is involved in the regulation of female receptivity is uncertain (*Deutsch et al., 2020*; *Schretter et al., 2020*; *Taisz et al., 2023*). However, when reducing EcR-A in pC1d neurons alone using the specific split-Gal4 SS56987 (*Schretter et al., 2020*), virgin female receptivity including copulation rate and latency to copulation did not change significantly compared with controls (*Figure 5—figure supplement 4*). These results suggested that the function of EcR-A in pC1b neurons is necessary for virgin female copulation rate.

As recently mated females may reduce sexual receptivity and increase egg laying (*Avila et al., 2011*; *Kubli, 2003*). we asked whether the decreased copulation rate induced by EcR-A could be a post-mating response and correlate with elevated egg laying. To address this, we examined the number of eggs laid by virgin females when EcR-A was reduced in pC1 neurons. We found that manipulation of EcR-A did not enhance egg laying significantly in virgin females (*Figure 5F*), although the UAS-EcR-A-RNAi control causes a massive decrease in female fertility. Meanwhile, we further analyzed whether reduction of EcR-A in pC1 neurons regulates the VPO or the OE. We found that reducing the EcR-A expression in pC1 neurons lead to the significantly less VPO and more OE (*Figure 5G, H*). These results suggested that reduced EcR-A expression in pC1 neurons results in the similar phenotype to that of mated females.

## EcR-A participates in the morphological development of pC1 neurons

EcR isoforms have distinct temporal and spatial expression patterns in the CNS (*Riddiford et al., 2000*; *Truman et al., 1994*). It is unknown when EcR-A functions in pC1 neurons for virgin female receptivity. Thus, we examined virgin female receptivity when EcR-A expression was conditionally reduced through RNAi via the pC1-ss2-Gal4 under the control of the temporal and regional gene expression targeting system (*McGuire et al., 2004*). EcR-A was reduced during the whole larval, pupal and adult stage, respectively (*Figure 5I–K*). When the experiment was done at the pupal stage is the only situation when the controls were both different from the experimental (*Figure 5J*). The result suggested that EcR-A in pC1 neurons plays a role in virgin female receptivity during metamorphosis. This is consistent with that PTTH regulates virgin female receptivity before the start of metamorphosis.

We then tested how EcR-A functions in pC1 neurons to modulate virgin female receptivity. First, we tested the morphology of pC1 neurons when reducing the expression of EcR-A in pC1 neurons. We found that the morphology of pC1-ss2-Gal4 expressing neurons appeared after the formation of

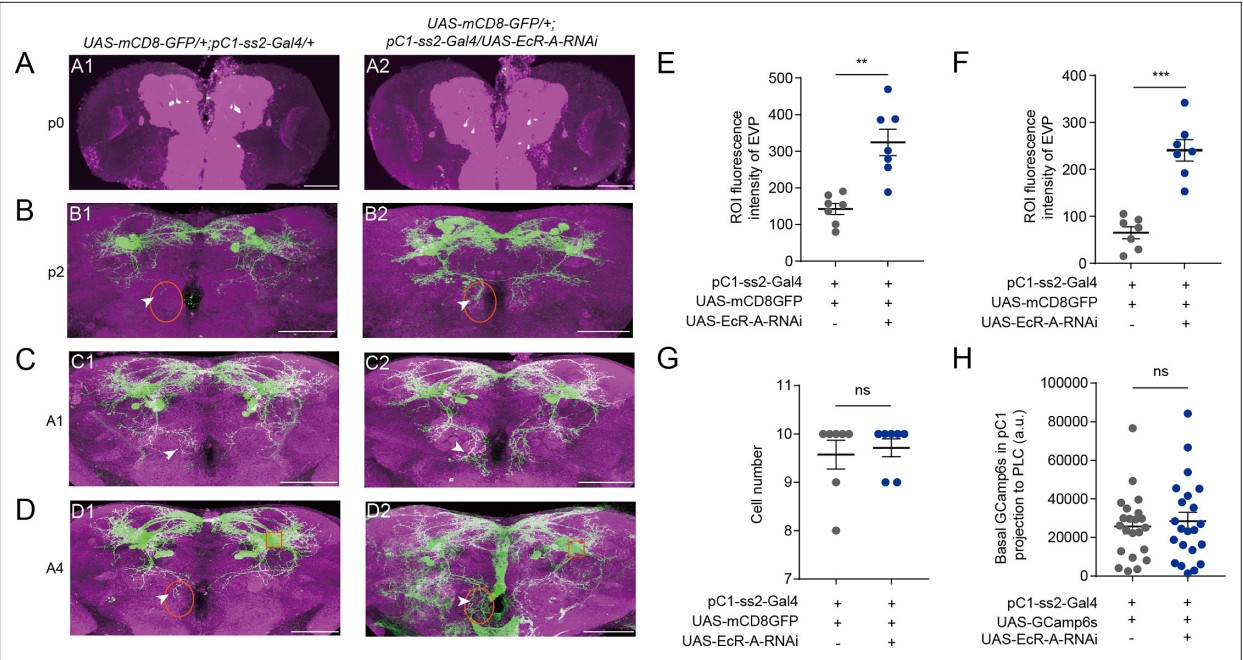

**Figure 6.** Reduced EcR-A in pC1 neurons induces the morphological changes. (**A1, A2**) pC1-ss2-Gal4 expressing neurons appeared at the start of the pupal stage. (**B–D**) Reduced EcR-A in pC1 neurons induced more elaborated morphologies of pC1d axons, especially the extra vertical projection (EVP). The EVP regions of pC1d neurons was indicated by arrows. The morphological changes appeared on the second day of the pupal stage (**B1, B2**) and retained to the adult stage including the first day (**C1, C2**) and the fourth day (**D1, D2**) of the adult stage. p0, the first day of the pupal stage; p2, the second day of the pupal stage; A1, the first day of the adult stage; A4, the fourth day of the adult stage. Fluorescence intensity of EVP in pC1d neurons on the second day of the pupal stage (**E**) and the fourth day of the adult stage (**F**) was quantified when EcR-A was reduced in pC1 neurons ($n = 7$). The quantified EVP regions were marked in (**B**) and (**D**) with orange ellipses. (**G**) pC1 neurons of the fourth day adults had comparable cell body number when EcR-A was reduced in pC1 neurons or not ($n = 7$). (**H**) Basal GCaMP6s signals in the lateral protocerebral complex (LPC) region of pC1 neurons when EcR-A was reduced in pC1 neurons ($n = 22$). LPC regions, the neurites extending from pC1 cell bodies, were marked with orange square in (**D1**) and (**D2**). The comparison referred to flies with decreased EcR isoform in pC1 neurons. Female flies in (**G, H**) were 4-day-old adults. Scale bars are 50 μm. For all comparisons, Mann–Whitney $U$ test is applied. Error bars indicate SEM. **$p < 0.01$, ***$p < 0.001$, ns indicates no significant difference.

The online version of this article includes the following source data for figure 6:

**Source data 1.** Fluorescence intensity, cell number, and calcium activity.

the white pupa (*Figure 6A1*). The reduced EcR-A expression induced the more elaborated morphologies of the pC1-d/e cells, especially the extra vertical projection (EVP) near the midline of brains (*Figure 6B–D*; *Deutsch et al., 2020*). These changes exhibited from the second day of the pupal stage (*Figure 6B1, B2, E*) and maintained at the adult stage (*Figure 6D1, D2, F*). Meanwhile, the number of pC1 cell bodies in adult flies when EcR-A was reduced were the same as that of wild-type flies (*Figure 6G*). Previous studies suggested that pC1d cells serve as a hub within the central brain for dsx[+] and fru[+] neurons (*Deutsch et al., 2020*). Thus, the abnormal development of pC1d neurons may induce the changes between pC1d neurons and other dsx[+] and fru[+] neurons to affect associated behaviors.

Furthermore, we asked whether reduced female copulation rate was due to that EcR-A expression affected the activity of pC1 neurons. Because all pC1 cells characterized so far project to the lateral junction of the lateral protocerebral complex (LPC) (*Kimura et al., 2015*; *Rezával et al., 2016*; *Scheffer et al., 2020*; *Wang et al., 2020b*; *Wu et al., 2019*; *Zhou et al., 2014*), we expressed GCamp6s in all pC1 neurons and tested the calcium signals in the lateral junction of LPC when EcR-A was knocked down (*Figure 6D1, D2*). Reduced EcR-A did not induce significantly different calcium responses in the LPC (*Figure 6H*). Thus, our results suggested that the decreased female copulation rate induced by reduced EcR-A in pC1 neurons was mainly due to the morphological changes of pC1b neurons, which then modulate the connections of pC1b neurons with other neurons.

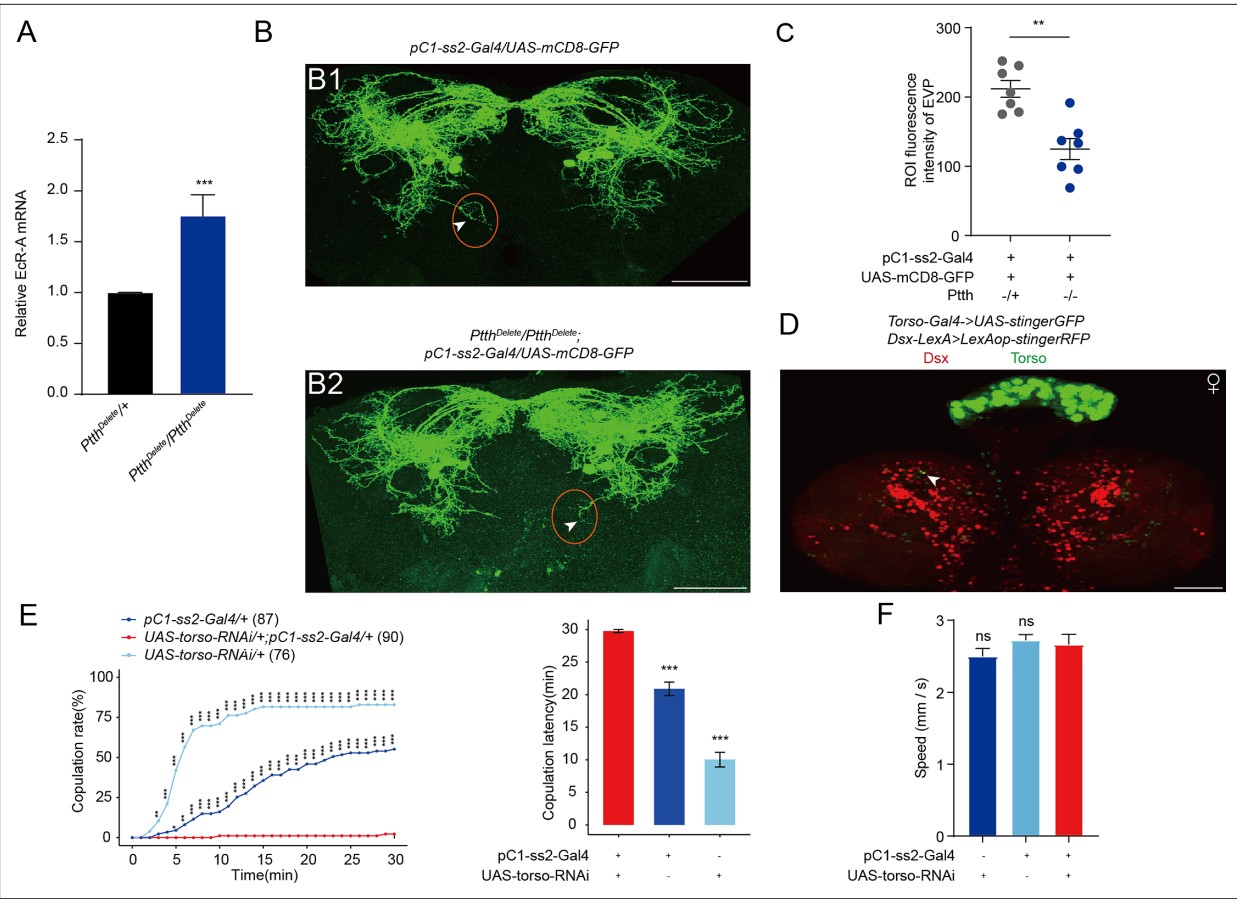

**Figure 7.** The function of prothoracicotropic hormone (PTTH) on EcR-A and pC1 neurons. (**A**) qRT-PCR for EcR-A when PTTH was deleted. Bars represent mean ± SEM. p values are from Mann–Whitney *U* test (*n* = 8 for *Ptth*<sup>Delete</sup>/+ and *n* = 6 for *Ptth*<sup>Delete</sup>/*Ptth*<sup>Delete</sup>, each sample contains about 10 bodies of newly formed prepupae). *p < 0.05, **p < 0.01, ***p < 0.001, ****p < 0.0001, ns indicates no significant difference. (**B**) Deletion of PTTH induced less elaborated morphologies of pC1d axons, especially the extra vertical projection (EVP). The EVP regions of pC1d neurons was indicated by arrows. Female flies were 4-day-old adults. (**C**) Fluorescence intensity of EVP in pC1d neurons on the fourth day was quantified when PTTH was deleted (*n* = 7). The quantified EVP regions were marked in (**B1**) and (**B2**) with red ellipses. (**D**) Torso-Gal4 and Dsx-LexA were labeled by stinger GFP and stingerRFP, respectively. Arrows indicated the overlap of GFP and RFP signals. Representative of five female brains. Scale bars, 50 μm. (**E, F**) Knock-down of Torso in pC1 neurons inhibited virgin female copulation rate and enhanced latency to copulation. Females had similar locomotion speeds between groups (**F**). The pC1-ss2-Gal4 control causes a massive decrease in female receptivity. The comparison referred to pC1-ss2-Gal4/UAS-torso-RNAi. Female flies were 4- to 6-day-old adults. The number of female flies paired with wild-type males is displayed in parentheses. For the copulation rate, chi-square test is applied. For the latency to copulation, Kruskal–Wallis ANOVA and post hoc Mann–Whitney *U* tests are applied. Error bars indicate SEM, *p < 0.05, **p < 0.01, ***p < 0.001, ns indicates no significant difference.

The online version of this article includes the following source data for figure 7:

**Source data 1.** Relative mRNA level, fluorescence intensity, copulation time, and walking speed.

## The function of PTTH on EcR-A and pC1 neurons

As newly formed prepupae, the ptth-Gal4 > UAS-Grim flies display similar changes in gene expression to the genetic control flies to response to a high-titer ecdysone pulse. These genes include the repression of EcR (*McBrayer et al., 2007*). According to the contradictory functions of PTTH deletion and EcR-A reduction in pC1 neurons, we wanted to know whether there is a similar feedforward relationship between PTTH and EcR-A. We quantified the EcR-A expression in the whole pupa body during the start of prepupa stage, when the pC1-ss2-Gal4 expressing neurons appear. Indeed, PTTH−/− induced upregulated EcR-A compared with PTTH−/+ flies (*Figure 7A*). This suggested the feedforward relationship between PTTH and EcR-A expression during the start of prepupa stage. Consistent with this, when PTTH was deleted, pC1 neurons exhibited the contradictory pattern to that when EcR-A was decreased in pC1 neurons (*Figure 7B, C*). These suggested the feedforward

relationship between PTTH and EcR-A, and may explain the contradictory functions of PTTH deletion and EcR-A reduction in pC1 neurons for female receptivity.

Furthermore, PTTH neurons are *dsx*-positive, almost all neurons regulating female receptivity express Dsx$^F$. We wanted to know whether PTTH neurons affect other dsx$^+$ neurons, including pC1 neurons. Indeed, we detected the slightly overlap of dsx-LexA>LexAop-RFP and torso-Gal4>UAS-GFP during larval stage (*Figure 7D*). Furthermore, decreasing torso expression in pC1 neurons significantly inhibit female receptivity (*Figure 7E*). The inhibited virgin female receptivity had no relationship either with the locomotion activity of virgin females (*Figure 7F*). These results suggest that, PTTH regulates female receptivity not only through ecdysone, but also may through regulating other neurons especially Dsx$^F$-positive neurons associated with female receptivity directly.

## Discussion

In this study, we found that peptide hormone PTTH negatively modulates virgin female receptivity through ecdysone. PTTH neurons are doublesex-positive and regulate virgin female receptivity during neural development. PTTH deletion resulted in the increased ecdysone receptor EcR-A expression in newly formed prepupae. Furthermore, EcR-A functions in pC1 neurons to positively regulate virgin female receptivity during metamorphosis mainly through modulating the anatomical morphology of pC1b neurons. Additionally, decreasing the expression of PTTH receptor torso in pC1 neurons inhibited female receptivity. Taken together, our results revealed the contrary functions of PTTH deletion and reduction of EcR-A in pC1 neurons during neurodevelopment. In addition, EcR-A in pC1 neurons regulates virgin female copulation rate during metamorphosis mainly through modulating the morphology of pC1b neurons.

Most of neurons regulating sexual behaviors in female flies are dsx-positive. Our results showed that PTTH neurons are also dsx$^+$ neurons. This suggested that PTTH neurons have relationships with other dsx$^+$ neurons and the juvenile–adult transition is regulated by *doublesex* gene. Indeed, we detected the overlap between torso-Gal4 signal and dsx-LexA signal at the larval stage. Furthermore, when PTTH receptor torso was decreased in dsx$^+$ pC1 neurons, female receptivity was inhibited. This suggested that PTTH functions in pC1 neurons through neuronal projection or endocrine pathway directly to regulate female receptivity. However, we did not detect the change of female receptivity when Dsx$^F$ was decreased in PTTH neurons. This suggested that Dsx$^F$ functions in PTTH neurons on other aspects, such as the development of PTTH neurons or the synthesis and release of PTTH to regulate development.

PTTH regulates virgin female receptivity in an ecdysone-dependent manner before metamorphosis. Ecdysone functions through its receptor EcR which is involved in all phases of the nervous system development. In our study, reduced EcR-A expression in all pC1 neurons, which encode the mating status of females, lead to the decreased copulation rate. While, reduced EcR-A in pC1-a, c and e simultaneously did not reduce the copulation rate significantly. This suggested that EcR-A plays the critical role in pC1-b and/or -d neurons for regulating virgin female receptivity. Our results revealed that reduced EcR-A induced the more elaborated morphologies of pC1d neurons. Previous studies detected the synaptic connections between the axons of pC1d and the dendrites of DNp13 neurons (*Deutsch et al., 2020*; *Mezzera et al., 2020*). DNp13 neurons are command neurons for OE. When females extruded, the ovipositor physically impedes copulation (*Mezzera et al., 2020*; *Wang et al., 2020a*). However, reduced EcR-A expression in pC1d neurons did not affect virgin female receptivity (*Figure 5—figure supplement 4*). This might be due to three possibilities. First, the more elaborated morphologies of pC1d neurons did not affect synaptic connections between pC1d and DNp13 neurons. Second, the unchanged pC1 neural activity could not affect the neural activity of DNp13 neurons. Third, the morphological change of pC1d neurons is not sufficient for the decreased copulation rate. To sum, these suggest that the morphological change of pC1b neurons is necessary for the decreased copulation rate. However, due to the lack of pC1b drivers, we could not rule out morphological changes in pC1b neurons when EcR-A was reduced.

In our study, PTTH negatively regulates female receptivity, while EcR-A positively regulates female receptivity. Previous study revealed that, expression of EcR is repressed in newly formed prepupae to response to high-titer ecdysone pulse (*McBrayer et al., 2007*). We detected the similar feedforward relationship between PTTH and EcR-A in newly formed prepupae. Consistent with this, we detected the contrary pattern of pC1d neurons when PTTH was deleted, compare with that when EcR-A was

decreased in pC1 neurons. However, it is not sure that PTTH deletion could result in the increased expression of EcR-A in pC1 neurons. In addition, PTTH deletion must affect the development of almost other neurons, but not only pC1 neurons. This maybe the reason for that reduction of Torso in pC1 neurons had contrary effect on female receptivity to that when PTTH is deleted. So, the feed-forward relationship between PTTH and EcR-A in newly formed prepupae is one possible reason for the contrary functions of PTTH deletion and reduction of EcR-A in pC1 neurons on female receptivity.

Previous studies have demonstrated that the development of fru⁺ neurons need EcR-A in male *D. melanogaster*. Furthermore, reduced EcR-A in fru⁺ neurons induced the male–male courtship (*Dalton et al., 2009*). We also detected the male–male courtship behavior when PTTH was deleted (data not shown) as previous study (*McBrayer et al., 2007*). This remits to the impact of ecdysone on dsx⁺ or fru⁺ neurons. Similarly, PTTH deletion may also affect the development of other neurons and further other behaviors such as the female fecundity and the body size (*McBrayer et al., 2007*; *Rewitz et al., 2009*; *Shimell et al., 2018*), although there is no sufficient evidence for the effects of fecundity and the body size on female receptivity. So, we could not exclude all effects of other aspects on female receptivity.

Our results suggested a regulatory role of PTTH in virgin female receptivity. Even though insects and mammals represent highly diverged classes, insects have evolved a similar strategy for triggering the juvenile–adult transition (*Herbison, 2016*; *Pan and O'Connor, 2019*). The juvenile–adult transition involves the HPG axis in mammals and the PG axis in insects. Among the neurons belonging to the axis, PTTH neurons and GnRH neurons have the similar function to stimulate the PG gland and pituitary gland to release hormones which trigger maturation, respectively. It will be interesting to study the function of GnRH neurons on the mammal sexual behaviors.

This work extends the understanding of how neurodevelopmental processes regulate adult sexual behavior.

# Materials and methods

## Key resources table

| Reagent type (species) or resource | Designation | Source or reference | Identifiers | Additional information |
|---|---|---|---|---|
| Antibody | Anti-Bruchpilot (nc82), mouse monoclonal | Developmental Studies Hybridoma Bank | Cat# nc82, RRID: AB_2314866 | IHC (1:40) |
| Antibody | Anti-*Drosophila* ecdysone receptor (EcR-A), mouse monoclonal | Developmental Studies Hybridoma Bank | Cat# 15G1a (EcR-A), RRID: AB_528214 | IHC (1:10) |
| Antibody | Anti-*Drosophila* ecdysone receptor (EcR-B1), mouse monoclonal | Developmental Studies Hybridoma Bank | Cat# AD4.4(EcR-B1), RRID: AB_2154902 | IHC (1:10) |
| Antibody | Anti-GFP, rabbit polyclonal | Thermo Fisher Scientific | Cat# A-11122, RRID: AB_221569 | IHC (1:1000) |
| Antibody | Anti-GFP, chicken polyclonal | Thermo Fisher Scientific | Cat# A10262, RRID: AB_2534023 | IHC (1:1000) |
| Antibody | Alexa Fluor 488, goat anti-rabbit polyclonal | Thermo Fisher Scientific | Cat# A-11034, RRID: AB_2576217 | IHC (1:500) |
| Antibody | Alexa Fluor 488, goat anti-chickent polyclonal | Thermo Fisher Scientific | Cat# A-11039; RRID: AB_2534096 | IHC (1:500) |
| Antibody | Alexa Fluor 488, goat anti-mouse polyclonal | Thermo Fisher Scientific | Cat# A-11029, RRID: AB_2534088 | IHC (1:500) |
| Antibody | Alexa Fluor 546, goat anti-rabbit polyclonal | Thermo Fisher Scientific | Cat# A-11010, RRID: AB_2534077 | IHC (1:500) |
| Antibody | Anti-RFP, rabbit polyclonal | Thermo Fisher Scientific | Cat# R10367, RRID: AB_10563941 | IHC (1:500) |
| Antibody | Alexa Fluor 647, goat anti-mouse polyclonal | Thermo Fisher Scientific | Cat# A-21235, RRID: AB_2535804 | IHC (1:500) |
| Antibody | Anti-PTTH, rabbit polyclonal | Zhou Lab, Chinese Academy of Sciences, this paper | N/A | IHC (1:1300) |
| Chemical compound, drug | Paraformaldehyde (PFA) | Electron Microscopy Sciences | Cat#15713 | 8% PFA diluted in 1× PBS at 1:4 or 1:2 |
| Chemical compound, drug | DPX Mountant | Sigma-Aldrich | Cat# 44581 | |
| Chemical compound, drug | Normal goat serum | Sigma-Aldrich | Cat# G9023 | |

*Continued on next page*

*Continued*

| Reagent type (species) or resource | Designation | Source or reference | Identifiers | Additional information |
|---|---|---|---|---|
| Chemical compound, drug | 20-Hydroxyecdysone | Cayman | Cat# 16145 | Dissolved in 95% ethanol, 0.2 mg/ml |
| Chemical compound, drug | TRIzol | Ambion | Cat# 15596018 | |
| Genetic reagent (*D. melanogaster*) | *LexAop2-mCD8::GFP* | Bloomington Stock Center | BL# 32203, RRID:BDSC_32203 | |
| Genetic reagent (*D. melanogaster*) | *;;UAS-mCD8::GFP* | Bloomington Stock Center | BL# 32194, RRID:BDSC_32194 | |
| Genetic reagent (*D. melanogaster*) | *;UAS-mCD8::GFP;* | Bloomington Stock Center | BL# 5137, RRID:BDSC_5137 | |
| Genetic reagent (*D. melanogaster*) | UAS-dTrpA1/cyo | Garrity Lab, Brandeis University | N/A | |
| Genetic reagent (*D. melanogaster*) | UAS-Kir2.1 | Bloomington Stock Center | BL# 6595, RRID:BDSC_6595 | |
| Genetic reagent (*D. melanogaster*) | Ptth-Gal4 | Rao Lab, Peking University | N/A | |
| Genetic reagent (*D. melanogaster*) | PtthLexA | Rao Lab, Peking University | N/A | |
| Genetic reagent (*D. melanogaster*) | ΔPTTH | Rao Lab, Peking University | N/A | |
| Genetic reagent (*D. melanogaster*) | UAS-PTTH | Zhou Lab, Chinese Academy of Sciences, this paper | N/A | |
| Genetic reagent (*D. melanogaster*) | isoCS | Rao Lab, Peking University | N/A | |
| Genetic reagent (*D. melanogaster*) | *elav-Gal4* | Rao Lab, Peking University | N/A | |
| Genetic reagent (*D. melanogaster*) | *UAS-GFPStinger* | Janelia Research Campus | N/A | |
| Genetic reagent (*D. melanogaster*) | *LexAop-tomato* | Janelia Research Campus | N/A | |
| Genetic reagent (*D. melanogaster*) | *LexAop2-FlpL* | Janelia Research Campus | N/A | |
| Genetic reagent (*D. melanogaster*) | *UAS >stop > mCD8-GFP* | Janelia Research Campus | N/A | |
| Genetic reagent (*D. melanogaster*) | *Dsx-Gal4* | Janelia Research Campus | N/A | |
| Genetic reagent (*D. melanogaster*) | *Dsx-LexA* | Janelia Research Campus | N/A | |
| Genetic reagent (*D. melanogaster*) | *tub-Gal80ts* | Pan Lab, Southeast University | BL# 7018, RRID:BDSC_7018 | |
| Genetic reagent (*D. melanogaster*) | *pC1-ss1-Gal4* | Wang Lab, Lingang Laboratory | N/A | |
| Genetic reagent (*D. melanogaster*) | *pC1-ss2- Gal4* | Wang Lab, Lingang Laboratory | N/A | |
| Genetic reagent (*D. melanogaster*) | *vpoDN-ss1-Gal4* | Wang Lab, Lingang Laboratory | N/A | |
| Genetic reagent (*D. melanogaster*) | *vpoDN-ss2-Gal4* | Wang Lab, Lingang Laboratory | N/A | |
| Genetic reagent (*D. melanogaster*) | *vpoDN-ss3-Gal4* | Wang Lab, Lingang Laboratory | N/A | |
| Genetic reagent (*D. melanogaster*) | *UAS-EcR-RNAi* | Bloomington Stock Center | BL# 9327, RRID:BDSC_9327 | |
| Genetic reagent (*D. melanogaster*) | *UAS-EcR-A-RNAi* | Bloomington Stock Center | BL# 9328, RRID:BDSC_9328 | |

*Continued on next page*

*Continued*

| Reagent type (species) or resource | Designation | Source or reference | Identifiers | Additional information |
|---|---|---|---|---|
| Genetic reagent (*D. melanogaster*) | *UAS-EcR-B1-RNAi* | Bloomington Stock Center | BL# 9329, RRID:BDSC_9329 | |
| Genetic reagent (*D. melanogaster*) | *pC1d-Gal4* | Bloomington Stock Center | BL# 86847, RRID:BDSC_86847 | |
| Genetic reagent (*D. melanogaster*) | *UAS-PTTH-RNAi* | VDRC | V102043 | |
| Genetic reagent (*D. melanogaster*) | *UAS-Dsx$^F$-RNAi* | Pan Lab | N/A | |
| Genetic reagent (*D. melanogaster*) | *UAS-Torso-RNAi* | Liu Lab | BL# 33627, RRID:BDSC_33627 | |
| Recombinant DNA reagent | pBSK-attP-3P3-RFP-loxP | *Deng et al., 2019* | N/A | |
| Recombinant DNA reagent | pBSK-attB-loxP-myc-T2A-Gal4Gal4-GMR-miniwhite | *Deng et al., 2019* | N/A | |
| Recombinant DNA reagent | pBSK-attB-loxP-V5-T2A-LexA::p65-GMR-miniwhite | *Deng et al., 2019* | N/A | |
| Software, algorithm | MATLAB | MathWorks, Natick, MA | https://www.mathworks.com/products/matlab.html | |
| Software, algorithm | ImageJ | National Institutes of Health | https://imagej.nih.gov/ij/ | |
| Software, algorithm | Prism 7 | GraphPad | https://www.graphpad.com/ | |
| Software, algorithm | R 4.1.3 | RStudio | https://www.r-project.org | |

## Fly stocks

Flies were reared on standard cornmeal-yeast medium under a 12-hr:12-hr dark:light cycle at 25°C and 60% humidity. All the knock-out lines in this study for screening have been published (*Pavlou and Goodwin, 2013*). The following strains were obtained from Dr. Yi Rao: *isoCS* (wild-type), *ΔPtth*, *Ptth-Gal4*, *Ptth-LexA*, *elav-Gal4*, and *UAS-Kir2.1* (BL#6595). UAS-dTrpA1 was a gift from Dr. Paul Garrity. *UAS-GFPStinger*, *LexAop-tomato*, *LexAop2-FlpL*, *UAS > stop > mCD8-GFP*, *dsx-Gal4*, and *dsx-LexA Mellert et al., 2010* have been described previously (*Pfeiffer et al., 2008*; *Pfeiffer et al., 2010*) and are obtained from Janelia Research Campus. *tub-Gal80$^{ts}$* (BL#7018) was provided by Dr. Yufeng Pan. *pC1-ss1-Gal4*, *pC1-ss2-Gal4*, *vpoDN-ss1-Gal4*, *vpoDN-ss2-Gal4*, and *vpoDN-ss3-Gal4* were provided by Dr. Kaiyu Wang. *Torso-RNAi* (BL# 33627) was a gift from Suning Liu. The following lines were obtained from the Bloomington *Drosophila* Stock Center: *UAS-EcR-RNAi* (BL# 9327), *UAS-EcR-A-RNAi* (BL# 9328), *UAS-EcR-B1-RNAi* (BL# 9329), *UAS-mCD8::GFP* (BL# 32194), *LexAop2-mCD8::GFP* (BL# 32203), *UAS-mCD8::GFP* (BL# 5137), and *pC1d-Gal4* (BL# 86847). *UAS-PTTH-RNAi* (v102043) was from Vienna *Drosophila* Resource Center (VDRC).

## Behavioral assays

Flies were reared at 25°C and 60% humidity under a 12-hr light:12-hr dark cycle. Virgin females and wild-type males were collected upon eclosion, placed in groups of 12 flies each and aged 4–6 days (except for the assays for PTTH mutant on different days after eclosion, and the molecular rescue assay for the 24-hr-old females) before carrying out behavioral assay except for the transient thermogenetic experiments. Female receptivity assays were conducted as previously described (*Wang et al., 2022*; *Zhou et al., 2014*). A virgin female of defined genotype and a wild-type male were gently cold anesthetized and, respectively, introduced into two layers of the courtship chambers separated by a removable transparent film. The flies were allowed to recover for at least 45 min before the film was removed to allow the pair of flies to contact. The mating behavior was recorded using a camera (Canon VIXIA HF R500) for 30 min at 17 fps for further analysis.

For transient activation experiment by dTrpA1 in adult stage, flies were reared at 23°C. Flies were loaded into courtship chamber and recovered for at least 30 min at 23°C, then were placed at 23°C (control group) or 29°C (experimental group) for 30 min prior to removing the film and videotaping. For activation experiment by dTrpA1 during development, flies were reared at 29°C during the specific

stages compared with the controls who were reared at 23°C all the time. Flies were loaded into court-ship chamber and recovered for at least 45 min at 23°C prior to removing the film and videotaping.

## Quantification and statistical analysis of female receptivity behavior

Two parameters including copulation rate and latency to copulation were used to characterize receptivity and we got the datasets of two parameters from the same flies. The time from removing the film to successful copulation was measured for each female. The number of females that had engaged in copulation by the end of each 1 min interval within 30 min were summed and plotted as a percentage of successful copulation. The latency to copulation was the time from removing the film to successful copulation. All the time points that female successfully copulated were manually analyzed and the data of unhealthy flies were discarded.

## Temporally restricted RNAi

tub-Gal80$^{ts}$ crosses were reared at either 18°C for control groups or 30°C for experimental groups. Virgin females were collected at eclosion and were placed in groups of 12 flies each and aged 4–6 days before carrying out behavior assay. Assays were tested at 23°C.

## Male courtship index

Courtship index was defined as the proportion of time the male followed, oriented toward and attempted to copulate the female within 5 min of courtship initiation, marked by the initial orientation toward and following the female.

## VPO and OE

A virgin female of defined genotype and a wild-type male were aspirated into the courtship chambers and, respectively, introduced into two layers of the courtship chambers separated by a removable transparent film. The flies were allowed to recover for 30 min before the film was removed. To allow visualization of VPO, we recorded uncompressed image sequences at 896 × 896 pixels and 50 frames per second using a Photron Mini AX camera (Photron) with an AF-S VR Micro-Nikkor 105 mm lens (Nikon). Instances of VPO and OE were scored blind to genotype from frame by-frame playback during the first 5 min of courtship or until copulation if it occurred within 5 min. Courtship initiation was defined as the male orienting toward and beginning to follow the female. Rare trials with fewer than 30 s of total courtship were discarded.

## Locomotion assays

The rearing and experimental conditions in locomotion assays were the same as that in the corresponding female receptivity assays, excepting that individual females were loaded in the chambers without males. Spontaneous movements of the flies were recorded with a camera (Canon VIXIA HF R500) for 30 min at 30 fps for further analysis. The activity of flies during the middle 10 min was analyzed to calculate the average walking speed using Ctrax software.

## Egg laying

Virgin females were collected upon eclosion and one fly was housed on standard medium at 25°C, 60% relative humidity, 12-hr light:12-hr dark and allowed to lay eggs in single vials. Each fly was transferred into new food tube every 24 hr. The number of eggs was counted at the end of each 24-hr period. The numbers during the third and fourth day were summed for statistics and plot.

## Immunohistochemistry

Whole brains of flies were dissected in 1× PBS (phosphate buffered saline) and fixed in 2% paraformaldehyde diluted in 1× PBS for 55 min at room temperature. The samples were washed with PBT (1× PBS containing 0.3% Triton X-100) for 1 hr (3 × 20 min), followed by blocking in 5% normal goat serum (Blocking solution, diluted in 0.3% PBT) for 1 hr at room temperature. Then, the samples were incubated in primary antibodies (diluted in blocking solution) for 18–24 hr at 4°C. Samples were washed with 0.3% PBT for 1 hr (3 × 20 min), then were incubated in secondary antibodies (diluted in blocking solution) for 18–24 hr at 4°C. Samples were washed with 0.3% PBT for 1 hr (3 × 20 min), then were fixed in 4% paraformaldehyde for 4 hr at room temperature. After washed with 0.3% PBT for 1 hr (3 ×

20 min), brains were mounted on poly-L-lysine-coated coverslip in 1× PBS. The coverslip was dipped for 5 min with ethanol of 30% → 50% → 70% → 95% → 100% sequentially at room temperature, and then dipped for 5 min three times with xylene. Finally, brains were mounted with DPX (Distyrene, Plasticizer and Xylene) and allowed DPX to dry for 2 days before imaging. Primary antibodies used were: chicken anti-GFP (1:1000; Life Technologies #A10262), rabbit anti-GFP (1:1000; Life Technologies #A11122), rabbit anti-RFP (1:1000; Life Technologies #R10367), rabbit anti-PTTH antibody (1:1300), mouse anti-nc82 (1:40; DSHB), mouse anti-EcR-A (1:10; AB_528214), and mouse anti-EcR-B1 (1:10; AB_2154902). Secondary antibodies used were: Alexa Fluor goat anti-chicken 488 (1:500; Life Technologies #A11039), Alexa Fluor goat anti-rabbit 488 (1:500; Life Technologies #A11034), Alexa Fluor goat anti-rabbit 546 (1:500; Life Technologies #A11010), Alexa Fluor goat anti-mouse 647 (1:500; Life Technologies #A21235), and Alexa Fluor goat anti-mouse 488 (1:500; Life Technologies #A11029).

## Confocal microscopy and image analysis

Confocal imaging was performed under an LSM 710 inverted confocal microscope (ZEISS, Germany), with a Plan-Apochromat 20×/0.8 M27 objective or an EC Plan-Neofluar 40×/1.30 oil DIC M27 objective, and later analyzed using Fiji software.

## Generation of anti-PTTH antibody

The antisera used to recognize PTTH peptide were raised in New Zealand white rabbits using the synthetic peptide N′-TSQSDHPYSWMNKDQPWQFKC-C′. The synthesis of antigen peptide, the production and purification of antiserum were performed by Beijing Genomics Institute (BGI).

## Generation of UAS-PTTH

pJFRC28-5XUAS-IVS-GFP-p10 (#12073; Fungene Biotechnology, Shanghai, China) was used for the generation of the pJFRC28-UAS-PTTH construct. The pJFRC28-10XUAS-IVS-GFP-p10 plasmid was digested with NotI and XbaI to remove the GFP coding sequence, and then the cDNA of PTTH was cloned into this plasmid by Gibson Assembly. The Kozak sequence was added right upstream of the ATG. UAS-PTTH constructs were injected and integrated into the attP2 site on the third chromosome through phiC31 integrase mediated transgenesis. The construct was confirmed using DNA sequencing through PCR. The primers used for cloning PTTH cDNA were as follows:

    UAS-PTTH-F
    ATTCTTATCCTTTACTTCAGGCGGCCGCAAAATGGATATAAAAGTATGGCGACTCC
    UAS-PTTH-R
    GTTATTTTAAAAACGATTCATTCTAGATCACTTTGTGCAGAAGCAGCCG

## Genomic DNA extraction and RT-PCR

Genomic DNA was extracted from 10 whole bodies of wandering flies using MightyPrep reagent for DNA (Takara #9182). Whole body RNA was extracted from 10 whole bodies of wandering flies using TRIzol (Ambion #15596018). cDNA was generated from total RNA using the Prime Script reagent kit (Takara #RR047A). Candidates of *ΔPtth* were characterized by the loss of DNA band in the deleted areas through PCR on the genomic DNA and cDNA. Primer sequences used in *Figure 1* are listed in *Supplementary file 1*.

## Measurements of pupariation timing and adult mass

The flies were reared at 25°C and 60% humidity under a 12-hr light:12-hr dark cycle. Two-hour time collections of embryos laid on standard food vials. Each vial contained 20–30 eggs. The range of time for pupariation was recorded for each vial. Sexed adults of 24-hr-old were weighted in groups of 10 flies using a NENVER-TP-214 microbalance at the same time.

## Identification of sex in *Drosophila* larvae

Third-instar larvae can be sexed (*True & John, 2014*). Gonads are translucent and visible in side view in the posterior third of the larva. The male gonads are about five times bigger than the female gonads. The identified wandering female and male larvae were reared in different vials for the subsequent experiments.

## Rescue by 20-hydroxyecdysone feeding

Thirty freshly ecdysed *ΔPtth* L3 larvae, grown at 25°C and 60% humidity under a 12-hr light:12-hr dark cycle, were washed with water and transferred to normal food for additional aging. After 20 hr, larvae were washed and transferred to a vial supplemented with either 20-hydroxyecdysone (20E, Cayman #16145, dissolved in 95% ethanol, final concentration 0.2 mg/ml) or 95% ethanol (same volume as 20E). The wild-type larvae were directly transferred to vials supplemented with 20E or 95% ethanol upon L3 ecdysis. Once seeded with L3 larvae, the vials were returned to 25°C and 60% humidity under a 12-hr light:12-hr dark cycle.

## Quantification of fluorescence intensity

The fluorescence intensity was quantified using Fiji software. The areas of interest (ROI) were marked in the slices including the interested regions and quantified using the 'plot z-axis profile' function. The fluorescence intensity in each slice was summed for statistics and plot. The parameters used for confocal imaging of each brain were the same.

## Calcium imaging

Flies aged 4–6 days were immobilized on ice for ~30 s. The brain was then dissected out in extracellular solution (ECS) that contains (in millimoles): 108 NaCl, 5 KC1, 5 trehalose, 5 sucrose, 5 HEPES (4-(2-Hydroxyethyl)piperazine-1-ethanesulfonic acid), 26 NaHCO$_3$, 1 NaH$_2$PO$_4$, 2 CaCl$_2$, and 1.5 MgCl$_2$ [pH 7.1–7.3 when bubbled with 95% (vol/vol) O$_2$/5% (vol/vol) CO$_2$, ~290 mOsm] and mounted on a poly-D-lysine coated coverslip. The samples were continuously perfused with ECS.

Calcium imaging was performed at 21°C on a customized two-photon microscope equipped with a resonant scanner (Nikon), a piezo objective scanner (Nikon) and a 40× water-immersion objective (Nikon). GCaMP6s was excited at 920 nm.

Analysis of calcium imaging data was done offline with NIS-Elements AR 5.30.01. Briefly, the square region of interest (ROIs), 25 pixels on the pC1 neurons in the center of lateral junction, was chosen for measurements. For each frame, the average fluorescence intensity of pixels within ROIs was calculated blind to genotype. The average fluorescence intensity of ROIs in each frame covering pC1 neurons was summed for statistics and plot.

## qRT-PCR

Total RNA was extracted from about 10 flies using TRIzol (Ambion #15596018). The cDNA was synthesized using Prime Script reagent kit (Takara #RR047A). Quantitative PCR was performed on Thermo Piko Real 96 (Thermo) using SYBR Green PCR Master Mix (Takara #RR820A). The mRNA expression level was calculated by the $2^{-\Delta\Delta Ct}$ method and the results were plotted by using tubulin as the reference gene. Primers are listed in *Supplementary file 1*. All reactions were performed in triplicate. The average of four biological replicates ± SEM was plotted.

## Statistical analysis

Statistical analyses were carried out using R software version 3.4.3 or Prism7 (GraphPad software). For the copulation rate, chi-square test is applied. The Mann–Whitney *U* test was applied for analyzing the significance of two columns. Kruskal–Wallis ANOVA test followed by post hoc Mann–Whitney *U* test was used to identify significant differences between multiple groups.

## Acknowledgements

We thank Yi Rao (Peking University), Yufeng Pan (Southeast University), Kaiyu Wang (Lingang Laboratory), Yan Zhu (Chinese Academy of Sciences), Paul Garrity (Brandeis University), Wei Zhang (Tsinghua University), Li Liu (Chinese Academy of Sciences), Suning Liu (South China Normal University), and Xuan Guo (Jinzhou Medical University), the Bloomington *Drosophila* Stock Center and Tsinghua Fly Center for sharing fly strains; Chenzhu Wang (Chinese Academy of Sciences), Yufeng Pan (Southeast University), Zhiqiang Yan (Shenzhen Bay Laboratory), and Yan Zhu (Chinese Academy of Sciences) for their comments; Xiangdong Li (Chinese Academy of Sciences), Fengming Wu (Chinese Academy of Sciences), Tao Wang (Chinese Academy of Sciences), and Jin Ge (Chinese Academy of Sciences) for

assistance with behavioral assays; Yihui Chen and Hongjiang Gao for the maintenance of materials; other members of the Zhou laboratory for helpful discussions.

## Additional information

### Funding

| Funder | Grant reference number | Author |
|---|---|---|
| Shenzhen Bay Laboratory | 21260061 | Chuan Zhou |
| Shenzhen Bay Laboratory | S239201006 | Jing Li |
| National Natural Science Foundation of China | Y711241133 | Chuan Zhou |
| Chinese Academy of Sciences | Y929731103 | Chuan Zhou |

The funders had no role in study design, data collection, and interpretation, or the decision to submit the work for publication.

### Author contributions

Jing Li, Conceptualization, Data curation, Software, Formal analysis, Supervision, Funding acquisition, Validation, Investigation, Visualization, Methodology, Writing – original draft, Project administration, Writing – review and editing; Chao Ning, Data curation, Software, Formal analysis, Investigation, Visualization, Methodology; Yaohua Liu, Ruixin Fang, Data curation, Formal analysis, Investigation; Bowen Deng, Resources; Bingcai Wang, Kai Shi, Rencong Wang, Software; Chuan Zhou, Conceptualization, Resources, Data curation, Software, Supervision, Funding acquisition, Validation, Investigation, Methodology, Project administration

### Author ORCIDs

Jing Li https://orcid.org/0000-0003-3248-3513

Reviewer #1 (Public Review): https://doi.org/10.7554/eLife.92545.3.sa1
Reviewer #2 (Public Review): https://doi.org/10.7554/eLife.92545.3.sa2
Reviewer #3 (Public Review): https://doi.org/10.7554/eLife.92545.3.sa3
Author response https://doi.org/10.7554/eLife.92545.3.sa4

## Additional files

### Supplementary files

• Supplementary file 1. The primers used for the verification of *Ptth*^*Delete*^ null mutant flies and for the real-time quantitative PCR of EcR-A when prothoracicotropic hormone (PTTH) is deleted.

• MDAR checklist

### Data availability

All study data are included in the main text and supporting information. This study does not involve new code. Fly stocks and reagents used in this study are available from the corresponding author upon reasonable request.

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
