## [Editor Report · eLife assessment]

The aim of this **valuable** study is to uncover developmental roles of the neuropeptide prothoracicotropic hormone (PTTH) and ecdysone, which later regulate female receptivity of *Drosophila melanogaster*. The work combines spatially and temporally restricted genetic manipulation with behavior quantification to explore these molecular pathways and the neuronal substrates participating in the control of female sexual receptivity. At present, the implication of both signaling pathways in this process is **convincing** but the strength of the evidence is **incomplete** to support the main claim that PTTH pathway controls female sexual receptivity through the function of ecdysone in pC1 neurons.

---

## [Referee Report · Reviewer #1 (Public Review)]

Summary

This article delves into the role of Ecdysone in regulating female sexual receptivity in *Drosophila*. The researchers discovered that PTTH, a positive regulator of Ecdysone production, hurts the receptivity of adult virgin females. Specifically, the researchers found that losing larval PTTH before metamorphosis significantly increases female receptivity immediately after adult eclosion. In addition, Ecdysone, through its receptor EcR-A, is necessary during metamorphic neurodevelopment for the proper development of P1 neurons, as its silencing leads to morphological changes associated with reduced adult female receptivity. Furthermore, Torso enhances receptivity in the adult stage. The molecular mechanisms linking each molecule to female receptivity have yet to be fully understood; therefore, the involvement of the juvenile-to-adult hormonal pathway (PTTH/Torso/ecdysone) in female receptivity is not proven.

Strengths

(1) Robust Methodology and Experimental Design: The study employs a comprehensive and well-structured experimental approach, combining genetic manipulations, behavioral assays, and molecular analyses. This multi-faceted methodology allows for a thorough investigation of the role of PTTH and Ecdysone in regulating female sexual receptivity in *Drosophila*. The use of specific gene knockouts, RNA interference, and overexpression techniques provides strong evidence supporting the findings.

(2) Clear and Substantial Findings: The authors provide compelling data showing that PTTH negatively regulates female receptivity during the larval stage, which is rescued by Ecdysone feeding. Instead, metamorphic Ecdysone has a positive role during neurodevelopment. The experiments demonstrate this dual and temporally distinct role of PTTH/Ecdysone, shedding light on a complex hormonal regulation mechanism.

(3) Clarification of Experimental Details: In response to the initial review, the authors have clarified important experimental details, such as the precise timing of genetic manipulations and the specific developmental stages examined. This clarification enhances the reproducibility and understanding of the study.

Weaknesses

(1) Unresolved Contradictions and Complexity in Results: Despite the detailed responses, the paper still presents complex and somewhat contradictory findings regarding the roles of PTTH, Torso, and Ecdysone. The observed increase in EcR-A expression in PTTH mutants and the nuanced explanation regarding the feedforward relationship, while insightful, do not fully resolve the initial confusion about the differing effects of PTTH and Ecdysone manipulations on female receptivity. This required more exploration.

(2) Insufficient Exploration of Mechanistic Pathways: The potential mechanisms underlying the role of PTTH/Torso-Ecdysone across different developmental stages remain underexplored. While the authors suggest a feedforward relationship and possible interaction with other neurons, these hypotheses are not thoroughly tested or elaborated upon, leaving gaps in the mechanistic understanding.

(3) Limited Scope of Validation Experiments: While the authors addressed some reviewer concerns about validation, the scope remains somewhat limited. The lack of existing PTTH mutants and the challenges in manipulating PTTH expression without affecting receptivity suggests that further work is needed to validate these pathways robustly. The inability to fully replicate the PTTHdelete phenotype through other means leaves some questions unanswered.

(4). Complexity in Interpretation of dsx-Positive Neurons: The relevance of dsx-positive neurons in the context of PTTH's effects on female receptivity remains ambiguous. Although the authors provide some context, the biological significance of these observations is not fully clarified.

Conclusion

The manuscript presents a well-conceived study with significant findings that advance the understanding of hormonal regulation of female receptivity in *Drosophila*. However, complexities in the data and unresolved mechanistic questions suggest that further work is needed to clarify the exact pathways and interactions involved. The authors' responses to feedback have strengthened the paper, but additional experiments and more thorough mechanistic exploration would enhance the robustness and clarity of the conclusions.

---

## [Referee Report · Reviewer #2 (Public Review)]

Summary:

The authors tried to identify novel adult functions of the classical *Drosophila* juvenile-adult transition axis (i.e. ptth-ecdysone). Surprisingly, larval ptth-expressing neurons expressed the sex-specific doublesex gene, thus belonging to the sexual dimorphic circuit. Lack of ptth during late larval development caused enhanced female sexual receptivity, effect rescued by supplying ecdysone in the food. Among many other cellular players, pC1 neurons control receptivity by encoding the mating status of females. Interestingly, during metamorphosis a subtype of pC1 neurons required Ecdysone Receptor A in order to regulate such female receptivity. A transcriptomic analysis using pC1-specific Ecdyone signaling down-regulation gives some hints of possible downstream mechanisms.

Strengths:

The manuscript showed solid genetic evidence that lack of ptth during development caused enhanced copulation rate in female flies, which includes ptth mutant rescue experiments by over-expressing ptth as well as by adding ecdysone-supplemented food. They also present elegant data dissecting the temporal requirements of ptth-expressing neurons by shifting animals from non-permissive to permissive temperatures, in order to inactivate neuronal function (although not exclusively ptth function). They showed that EcR-A is up-regulated in ptth mutant background. By combining different drivers together with EcR-A RNAi and torso RNAi lines authors also identified the Ecdysone receptor and torso requirements of a particular subtype of pC1 neurons during metamorphosis. Convincing live calcium imaging showed no apparent effect of EcR-A in neural activity, although some effect on morphology is uncovered. Finally, bulk RNAseq shows differential gene expression after EcR-A down-regulation.

Weaknesses:

The paper has three main weaknesses. The first one refers to temporal requirements of ptth and ecdysone signaling. Whereas ptth is necessary during larval development, ecdysone effect appears during pupal development. ptth induces ecdysone synthesis during larval development but there is no published evidence about a similar role for ptth during pupal stages. The down-regulation of EcR-A by RNAi requires at least 8 h to be complete, whereas the activation of ptth neurons in larva stages is immediate. Furthermore, larval and pupal ecdysone functions are different (triggering metamorphosis vs tissue remodeling). The second caveat is the fact that ptth and ecdysone/torso loss-of-function experiments render opposite effects (enhancing and decreasing copulation rates, respectively). The most plausible explanation is that both functions are independent of each other, also suggested by differential temporal requirements. Finally, in order to identify the effect in the transcriptional response of down-regulating EcR-A in a very small population of neurons, a scRNAseq study should have been performed instead of bulk RNAseq.

In summary, despite the authors providing convincing evidence that ptth and ecdysone signaling pathways are involved in female receptivity, the main claim that ptth regulates this process through ecdysone is not supported by results. More likely, they'd rather be independent processes.

---

## [Referee Report · Reviewer #3 (Public Review)]

Summary:

This manuscript shows that mutations that disable the gene encoding the PTTH gene cause an increase in female receptivity (they mate more quickly), a phenotype that can be reversed by feeding these mutants the molting hormone, 20-hydoxyecdysone (20E). The use of an inducible system reveals that inhibition or activation of PTTH neurons during the larval stages increases and decreases female receptivity, respectively, suggesting that PTTH is required during the larval stages to affect the receptivity of the (adult) female fly. Showing that these neurons express the sex-determining gene dsx leads the authors to show that interfering with 20E actions in pC1 neurons, which are dsx-positive neurons known to regulate female receptivity, reduces female receptivity and increases the arborization pattern of pC1 neurons. The work concludes by showing that targeted knockdown of EcRA in pC1 neurons causes 527 genes to be differentially expressed in the brains of female flies, of which 123 passed a false discovery rate cutoff of 0.01; interestingly, the gene showing the greatest down-regulation was the gene encoding dopamine beta-monooxygenase.

This reviewer appreciates the effort that was done to revise the manuscript and address the various comments made by the reviewers. Nevertheless, I feel that the main concerns remain. These are not necessarily due to an unwillingness on the part of the authors to address them, but rather to difficulties that are inherent to trying to assign specific roles to EcR and pC1 neurons at a time when major changes are occurring (or are about to occur) in the nervous system, and do so using tools that are currently not sharp or specific enough. Many of the conclusions are supported by the results and those that may have alternative interpretations can remain more speculative until better tools become available. It is, nevertheless, an interesting and provocative piece of work.

Strengths

This is an interesting piece of work, which may shed light on the basis for the observation noted previously that flies lacking PTTH neurons show reproductive defects ("... females show reduced fecundity"; McBrayer, 2007; DOI 10.1016/j.devcel.2007.11.003).

Weaknesses:

There are some results whose interpretation seem ambiguous and findings whose causal relationship is implied but not demonstrated.

(1) At some level, the findings reported here are not at all surprising. Since 20E regulates the profound changes that occur in the central nervous system (CNS) during metamorphosis, it is not surprising that PTTH would play a role in this process. Although animals lacking PTTH (rather paradoxically) live to adulthood, they do show greatly extended larval instars and a corresponding great delay in the 20E rise that signals the start of metamorphosis. For this reason, concluding that PTTH plays a SPECIFIC role in regulating female receptivity seems a little misleading, since the metamorphic remodeling of the entire CNS is likely altered in PTTH mutants. Since these mutants produce overall normal (albeit larger--due to their prolonged larval stages) adults, these alterations are likely to be subtle. Courtship has been reported as one defect expressed by animals lacking PTTH neurons, but this behavior may stand out because reduced fertility and increased male-male courtship (McBrayer, 2007) would be noticeable defects to researchers handling these flies. By contrast, detecting defects in other behaviors (e.g., optomotor responses, learning and memory, sleep, etc) would require closer examination. For this reason I would ask the authors to temper their statement that PTTH is SPECIFICALLY involved in regulating female receptivity.

(2) The link between PTTH and the role of pC1 neurons in regulating female receptivity is not clear. Again, since 20E controls the metamorphic changes that occur in the CNS, it is not surprising that 20E would regulate the arborization of pC1 neurons. And since these neurons have been implicated in female receptivity, it would therefore be expected that altering 20E signaling in pC1 neurons would affect this phenotype. However, this does not mean that the defects in female receptivity expressed by PTTH mutants are due to defects in pC1 arborization. For this the authors would at least have to show that PTTH mutants show the changes in pC1 arborization shown in Fig. 6. And even then the most that could be said is that the changes observed in these neurons "may contribute" to the observed behavioral changes. Indeed, the changes observed in female receptivity may be caused by PTTH/20E actions on different neurons.

(3) Some of the results need commenting on, or refining, or revising:

(a) For some assays PTTH behaves sometimes like a recessive gene and at other times like a semi-dominant, and yet at others like a dominant gene. For instance, in Fig. 1D-G, PTTH[-]/+ flies behave like wildtype (D), express an intermediate phenotype (E-F), or behave like the mutant (G). This may all be correct but merits some comment.

(b) Some of the conclusions are overstated. (i) Although Fig. 2E-G does show that silencing the PTTH neurons during the larval stages affects copulation rate (E) the strength of the conclusion is tempered by the behavior of one of the controls (tub-GAL80[ts]/+, UAS-Kir2.1/+) in panels F and G, where it behaves essentially the same as the experimental group (and quite differently from the PTTH-GAL4/+ control; blue line).(Incidentally, the corresponding copulation latency should also be shown for these data.). (ii) For Fig. 5I-K, the conclusion stated is that "Knock-down of EcR-A during pupal stage significantly decreased the copulation rate." Although strictly correct, the problem is that panel J is the only one for which the behavior of the control lacking the RNAi is not the same as that of the experimental group. Thus, it could just be that when the experiment was done at the pupal stage is the only situation when the controls were both different from the experimental. Again, the results shown in J are strictly speaking correct but the statement is too definitive given the behavior of one of the controls in panels I and K. Note also that panel F shows that the UAS-RNAi control causes a massive decrease in female fertility, yet no mention is made of this fact.

---

## [Author Response]

The following is the authors’ response to the original reviews.

**Reviewer #1 (Public Review):**
Summary: This article explores the role of Ecdysone in regulating female sexual receptivity in *Drosophila*. The researchers found that PTTH, throughout its role as a positive regulator of ecdysone production, negatively affects the receptivity of adult virgin females. Indeed, loss of larval PTTH before metamorphosis significantly increases female receptivity right after adult eclosion and also later. However, during metamorphic neurodevelopment, Ecdysone, primarily through its receptor EcR-A, is required to properly develop the P1 neurons since its silencing led to morphological changes associated with a reduction in adult female receptivity. Nonetheless, the result shown in this manuscript sheds light on how Ecdysone plays a dual role in female adult receptivity, inhibiting it during larval development and enhancing it during metamorphic development. Unfortunately, this dual and opposite effect in two temporally different developmental stages has not been highlighted or explained.Strengths: This paper exhibits multiple strengths in its approach, employing a well-structured experimental methodology that combines genetic manipulations, behavioral assays, and molecular analysis to explore the impact of Ecdysone on regulating virgin female receptivity in *Drosophila*. The study provides clear and substantial findings, highlighting that removing PTTH, a positive Ecdysone regulator, increases virgin female receptivity. Additionally, the research expands into the temporal necessity of PTTH and Ecdysone function during development.Weaknesses:There are two important caveats with the data that are reflecting a weakness:(1) Contradictory Effects of Ecdysone and PTTH: One notable weakness in the data is the contrasting effects observed between Ecdysone and its positive regulator PTTH. PTTH loss of function increases female receptivity, while ecdysone loss of function reduces it. Given that PTTH positively regulates Ecdysone, one would expect that the loss of function of both would result in a similar phenotype or at least a consistent directional change.

A1. As newly formed prepupae, the ptth-Gal4>UAS-Grim flies display similar changes in gene expression to the genetic control flies to response to a high-titer ecdysone pulse. These include the repression of EcR (McBrayer et al.,2007). We tested whether there is a similar feedforward relationship between PTTH and EcR-A. We quantified the EcR-A mRNA level of PTTH -/- and PTTH -/+ in the whole body of newly formed prepupae. Indeed, PTTH -/- induced increased EcR-A expression in the whole body of newly formed prepupae compared with PTTH -/+ flies. Because of the function of EcR-A in gene expression, this suggests that PTTH -/- disturbs the regulation of a serious of gene expressions during metamorphosis. However, it is not sure that the EcR-A expression in pC1 neurons is increased compared with genetic controls when PTTH is deleted. Furthermore, PTTH -/- must affect development of other neurons rather than only pC1 neurons. So, the feedforward relationship between PTTH and EcRA at the start of prepupal stage is one possible cause for the contradictory effects of PTTH -/- and EcR-A RNAi in pC1 neurons.

(2) Discordant Temporal Requirements for Ecdysone and PTTH: Another weakness lies in the different temporal requirements for Ecdysone and PTTH. The data from the manuscript suggest that PTTH is necessary during the larval stage, as shown in Figure 2 E-G, while Ecdysone is required during the pupal stage, as indicated in Figure 5 I-K. Ecdysone is a crucial developmental hormone with precisely regulated expression throughout development, exhibiting several peaks during both larval and pupal stages. PTTH is known to regulate Ecdysone during the larval stage, specifically by stimulating the kinetics of Ecdysone peaking at the wandering stage. However, it remains unclear whether pupal PTTH, expressed at higher levels during metamorphosis, can stimulate Ecdysone production during the pupal stage. Additionally, given the transient nature of the Ecdysone peak produced at wandering time, which disappears shortly before the end of the prepupal stage, it is challenging to infer that larval PTTH will regulate Ecdysone production during the pupal stage based on the current state of knowledge in the neuroendocrine field.Considering these two caveats, the results suggest that the authors are witnessing distinct temporal and directional effects of Ecdysone on virgin female receptivity.

A2. First of all, it is necessary to clarify the detailed time for the manipulation of *Ptth* gene and PTTH neurons. In Figure 3, activation of PTTH neurons during the stage 2 inhibited the female receptivity. The “stage 2” is from six hours before the 3rd-instar larvae to the end of the wandering larvae (the start of prepupae). In Figure 5, The “pupal stage” is from the prepupal stage to the end of pupal stage. This “pupal stage” includes the forming of prepupae when the ecdysone peak is not disappeared. The time of manipulating *Ptth* and EcR-A in pC1 neurons are continuous. In addition, the pC1-Gal4 expressing neurons appear also at the start of prepupal stage. So, it is possible that PTTH regulates female receptivity through the function of EcR-A in pC1 neurons.

**Reviewer #1 (Recommendations For The Authors):**
In light of the significant caveat previously discussed, I will just make a few general suggestions:(1) The paper primarily focuses on robust phenotypes, particularly in PTTH mutants, with a well-detailed execution of several experiments, resulting in thorough and robust outcomes. However, due to the caveat previously presented (opposite effect in larva and pupa), consider splitting the paper into two parts: Figures 1 to 4 deal with the negative effect of PTTH-Ecdysone on early virgin female receptivity, while Figures 5 to 7 focus on the positive metamorphic effect of Ecdysone in P1 metamorphic neurodevelopment. However, in this scenario, the mechanism by which PTTH loss of function increases female receptivity should be addressed.

A3. It is a good suggestion that splitting the paper into two parts associated with the PTTH function and EcR function in pC1 neurons separately, if it is impossible that PTTH functions in female receptivity through the function of EcR-A in pC1 neurons. However, because of the feedforward relationship between PTTH and EcR-A in the newly formed prepupae, and the time of manipulating *Ptth* and EcR-A in pC1 neurons is continuous, it is possible that these two functions are not independent of each other. So, we still keep the initial edition.

(2) Validate the PTTH mutants by examining homozygous mutant phenotypes and the dose-dependent heterozygous mutant phenotype using existing PTTH mutants. This could also be achieved using RNAi techniques.

A4. We did not get other existing PTTH mutants. We instead decreased the PTTH expression in PTTH neurons and dsx+ neurons, but did not detect the similar phenotype to that of PTTH -/-. Similarly, the overexpression through PTTH-Gal4>UAS-PTTH is also not sufficient to change female receptivity. It is possible that both decreasing and increasing PTTH expression are not sufficient to change female receptivity.

(3) Clarify if elav-Gal4 is not expressed in PTTH neurons and discuss how the rescue mechanisms work (hormonal, paracrine, etc.) in the text.

A5. We tested the overlap of elav-Gal4>GFP signal and the stained PTTH with PTTH antibody. We did not detect the overlap. It suggests that elav-Gal4 is not expressed in PTTH neurons. However, we detected the expression of PTTH (PTTH antibody) in CNS when overexpressed PTTH using elav-Gal4>UASPTTH based on PTTH -/-. Furthermore, this rescued the phenotype of PTTH -/- in female receptivity. Insect PTTH isoforms have similar probable signal peptide for secreting. Indeed, except for the projection of axons to PG gland, PTTH also carries endocrine function acting on its receptor Torso in light sensors to regulate light avoidance of larvae. The overexpressed PTTH in other neurons through elav-Gal4>UASPTTH may act on the PG gland through endocrine function and then induce the ecdysone synthesis and release. So that, although elav-Gal4 is not expressed in PTTH neurons, the ecdysone synthesis triggered by PTTH from the hemolymph may result in the rescued PTTH -/- phenotype in female receptivity.

(4) Consider renaming the new PTTH mutant to avoid confusion with the existing PTTHDelta allele.

A6. We have renamed our new PTTH mutant as *PtthDelete*.

(5) Include the age of virgin females in each figure legend, especially for Figures 2 to 7, to aid in interpretation. This is essential information since wild-type early virgins -day 1- show no receptivity. In contrast, they reach a typical 80% receptivity later, and the mechanism regulating the first face might differ from the one occurring later.

A7. We have included the age of virgin females in each figure legend.

(6) Explain the relevance of observing that PTTH adult neurons are dsx-positive, as it's unclear why this observation is significant, considering that these neurons are not responsible for the observed receptivity effect in virgin females. Alternatively, address this in the context of the third instar larva or clarify its relevance.

A8. We decreased the DsxF expression in PTTH neurons and did not detect significantly changed female receptivity. Almost all neurons regulating female receptivity, including pC1 neurons, express DsxF. We suppose that PTTH neurons have some relationship with other DsxF-positive neurons which regulate female receptivity. Indeed, we detected the overlap of dsx-LexA>LexAop-RFP and torso-Gal4>UAS-GFP during larval stage. Furthermore, decreasing Torso expression in pC1 neurons significantly inhibit female receptivity.

These results suggest that, PTTH regulates female receptivity not only through ecdysone, but also may through regulating other neurons especially DsxF-positive neurons associated with female receptivity directly.

**Reviewer #2 (Public Review):**
Summary: The authors tried to identify novel adult functions of the classical *Drosophila* juvenile-adult transition axis (i.e. ptth-ecdysone). Surprisingly, larval ptth-expressing neurons expressed the sex-specific doublesex gene, thus belonging to the sexual dimorphic circuit. Lack of ptth during late larval development caused enhanced female sexual receptivity, an effect rescued by supplying ecdysone in the food. Among many other cellular players, pC1 neurons control receptivity by encoding the mating status of females. Interestingly, during metamorphosis, a subtype of pC1 neurons required Ecdysone Receptor A in order to regulate such female receptivity. A transcriptomic analysis using pC1-specific Ecdyone signaling down-regulation gives some hints of possible downstream mechanisms.Strengths: the manuscript showed solid genetic evidence that lack of ptth during development caused enhanced copulation rate in female flies, which includes ptth mutant rescue experiments by overexpressing ptth as well as by adding ecdysone-supplemented food. They also present elegant data dissecting the temporal requirements of ptth-expressing neurons by shifting animals from non-permissive to permissive temperatures, in order to inactivate neuronal function (although not exclusively ptth function). By combining different drivers together with a EcR-A RNAi line authors also identified the Ecdysone receptor requirements of a particular subtype of pC1 neurons during metamorphosis. Convincing live calcium imaging showed no apparent effect of EcR-A in neural activity, although some effect on morphology is uncovered. Finally, bulk RNAseq shows differential gene expression after EcR-A down-regulation.Weaknesses: the paper has three main weaknesses. The first one refers to temporal requirements of ptth and ecdysone signaling. Whereas ptth is necessary during larval development, the ecdysone effect appears during pupal development. ptth induces ecdysone synthesis during larval development but there is no published evidence about a similar role for ptth during pupal stages. Furthermore, larval and pupal ecdysone functions are different (triggering metamorphosis vs tissue remodeling). The second caveat is the fact that ptth and ecdysone loss-of-function experiments render opposite effects (enhancing and decreasing copulation rates, respectively). The most plausible explanation is that both functions are independent of each other, also suggested by differential temporal requirements. Finally, in order to identify the effect in the transcriptional response of down-regulating EcR-A in a very small population of neurons, a scRNAseq study should have been performed instead of bulk RNAseq.In summary, despite the authors providing convincing evidence that ptth and ecdysone signaling pathways are involved in female receptivity, the main claim that ptth regulates this process through ecdysone is not supported by results. More likely, they'd rather be independent processes.

B1. Clarification: in Figure 3, activation of PTTH neurons during the stage 2 inhibited the female receptivity. The “stage 2” is from six hours before the 3rd-instar larvae to the end of the wandering larvae (the start of prepupae). In Figure 5, The “pupal stage” is from the start of prepupal stage to the end of pupal stage. This “pupal stage” includes the forming of prepupae when the ecdysone peak is not disappeared. The time of manipulating *Ptth* and EcR-A in pC1 neurons are continuous. In addition, the pC1-Gal4 expressing neurons appear also at the start of prepupal stage. So, it is possible that PTTH regulates female receptivity through the function of EcR-A in pC1 neurons.

B2. During the forming of prepupae, the ptth-Gal4>UAS-Grim flies display similar changes in gene expression to the genetic control flies to response to a high-titer ecdysone pulse. These include the repression of EcR (McBrayer et al.,2007). We tested whether there is a similar feedforward relationship between PTTH and EcR-A. We quantified the EcR-A mRNA level of PTTH -/- and PTTH -/+ in the whole body of newly formed prepupae. Indeed, PTTH -/- induced increased EcR-A compared with PTTH -/+ flies. Because of the function of EcR-A in gene expression, this suggests that PTTH -/- disturbs the regulation of a serious of gene expressions during metamorphosis. However, it is not sure that the EcR-A expression in pC1 neurons is increased compared with genetic controls when PTTH is deleted. Furthermore, PTTH -/- must affect the development of other neurons rather than only pC1 neurons. So, the feedforward relationship between PTTH and EcR-A at the start of prepupal stage is one possible cause for the contradictory effects of PTTH -/- and EcR-A RNAi in pC1 neurons.

B3. We will do single cell sequencing in pC1 neurons for the exploration of detailed molecular mechanism of female receptivity in the future.

Reviewer #2 (Recommendations For The Authors):Additional experiments and suggestions:- torso LOF in the PG to determine whether or not the ecdysone peak regulated by ptth (there is a 1-day delay in pupation) is responsible for the ptth effect in L3. In the same line, what happens if torso is downregulated in the pC1 neurons? Is there any effect on copulation rates?

B4. Because the loss of phm-Gal4, we could not test female receptivity when decreasing the expression of Torso in PG gland. However, decreasing Torso expression in pC1 neurons significantly inhibit female receptivity. This suggests that PTTH regulates female receptivity not only through ecdysone but also through regulating dsx+ pC1 neurons in female receptivity directly.

- What is the effect of down-regulating ptth in the dsx+ neurons? No ptth RNAi experiments are shown in the paper.

B5. We decreased PTTH expression in dsx+ neurons but did not detect the change in female receptivity. We also decreased PTTH expression in PTTH neurons using PTTH-Gal4, also did not detect the change in female receptivity. Similarly, the overexpression through PTTH-Gal4>UAS-PTTH is also not sufficient to change female receptivity. It is possible that both decreasing and increasing PTTH expression are not sufficient to change female receptivity.

- Why are most copulation rate experiments performed between 4-6 days after eclosion? ptth LOF effect only lasts until day 3 after eclosion (but very weak-fig 1). Again, this supports the idea that ptth and ecdysone effects are unrelated.

B6. Most behavioral experiments were performed between 4-6 days after eclosion as most other studies in flies, because the female receptivity reaches the peak at that time. Ptth LOF made female receptivity enhanced from the first day after eclosion. This seems like the precocious puberty. Wild type females reach high receptivity at 2 days after eclosion (about 75% within 10 min). We suppose that Ptth LOF effect only lasts until day 3 after eclosion because too high level of receptivity of control flies to exceed.

It is not sure whether the effect of PTTH-/- in female receptivity disappears after the 3rd day of adult flies. So that it is not sure whether PTTH and EcR-A effects in pC1 neurons are unrelated.

- The fact that pC1d neuronal morphology changes (and not pC1b) does not explain the effect of EcR-A LOF. Despite it is highlighted in the discussion, data do not support the hypothesis. How do these pC1 neurons look like in a ptth mutant animal regarding Calcium imaging and/or morphology?

B7. We detected the pattern of pC1 neurons when PTTH is deleted. Consistent with the feedforward relationship between PTTH and expression of EcR-A in newly formed prepupae, PTTH deletion induced less established pC1-d neurons contrary to that induced by EcR-A reduction in pC1 neurons. However, it is not sure that the expression of EcR-A in pC1 neurons is increased when PTTH is deleted. Furthermore, on the one hand, manipulation of PTTH has general effect on the neurodevelopment not only regulating pC1 neurons. On the other hand, the detailed pattern of pC1-b neurons which is the key subtype regulating female receptivity when EcR-A is decreased in pC1 neurons or PTTH is deleted could not be seen clearly. So, the abnormal development of pC1-b neurons, if this is true, is just one of the possible reasons for the effect of PTTH deletion on female receptivity.

- The discussion is incomplete, especially the link between ptth and ecdysone; discuss why the phenotype is the opposite (ptth as a negative regulator of ecdysone in the pupa, for instance); the difference in size due to ptth LOF might be related to differential copulation rates.

B8. We have revised the discussion. We could not exclude the effect of size of body on female receptivity when PTTH was deleted or PTTH neurons were manipulated, although there was not enough evidence for the effect of body size on female receptivity.

- scheme of pC neurons may help.

B9. We have tried to label pC1 neurons with GFP and sort pC1 neurons through flow cytometry sorting, but could not success. This may because the number of pC1 neurons is too low in one brain. We will try single-cell sequencing in the future.

- Immunofluorescence images are too small.

B10. We have resized the small images.

**Reviewer #3 (Public Review):**
Summary:This manuscript shows that mutations that disable the gene encoding the PTTH gene cause an increase in female receptivity (they mate more quickly), a phenotype that can be reversed by feeding these mutants the molting hormone, 20-hydoxyecdysone (20E). The use of an inducible system reveals that inhibition or activation of PTTH neurons during the larval stages increases and decreases female receptivity, respectively, suggesting that PTTH is required during the larval stages to affect the receptivity of the (adult) female fly. Showing that these neurons express the sex-determining gene dsx leads the authors to show that interfering with 20E actions in pC1 neurons, which are dsx-positive neurons known to regulate female receptivity, reduces female receptivity and increases the arborization pattern of pC1 neurons. The work concludes by showing that targeted knockdown of EcRA in pC1 neurons causes 527 genes to be differentially expressed in the brains of female flies, of which 123 passed a false discovery rate cutoff of 0.01; interestingly, the gene showing the greatest down-regulation was the gene encoding dopamine beta-monooxygenase.StrengthsThis is an interesting piece of work, which may shed light on the basis for the observation noted previously that flies lacking PTTH neurons show reproductive defects ("... females show reduced fecundity"; McBrayer, 2007; DOI 10.1016/j.devcel.2007.11.003).Weaknesses:There are some results whose interpretation seem ambiguous and findings whose causal relationship is implied but not demonstrated.(1) At some level, the findings reported here are not at all surprising. Since 20E regulates the profound changes that occur in the central nervous system (CNS) during metamorphosis, it is not surprising that PTTH would play a role in this process. Although animals lacking PTTH (rather paradoxically) live to adulthood, they do show greatly extended larval instars and a corresponding great delay in the 20E rise that signals the start of metamorphosis. For this reason, concluding that PTTH plays a SPECIFIC role in regulating female receptivity seems a little misleading, since the metamorphic remodeling of the entire CNS is likely altered in PTTH mutants. Since these mutants produce overall normal (albeit larger--due to their prolonged larval stages) adults, these alterations are likely to be subtle. Courtship has been reported as one defect expressed by animals lacking PTTH neurons, but this behavior may stand out because reduced fertility and increased male-male courtship (McBrayer, 2007) would be noticeable defects to researchers handling these flies. By contrast, detecting defects in other behaviors (e.g., optomotor responses, learning and memory, sleep, etc) would require closer examination. For this reason, I would ask the authors to temper their statement that PTTH is SPECIFICALLY involved in regulating female receptivity.

C1. We agree with that, it is not surprising that PTTH regulates the profound changes that occur in the CNS during metamorphosis through ecdysone. Also, the behavioral changes induced by PTTH mutants include not only female receptivity. We will temper the statement about the function of PTTH on female receptivity.

We think there are two new points in our text although more evidences are needed in the future. On the one hand, PTTH deletion and the reduction of EcR-A in pC1 neurons during metamorphosis have opposite effects on female receptivity. On the other hand, development of pC1-b neurons regulated by EcR-A during metamorphosis is important for female receptivity.

(2) The link between PTTH and the role of pC1 neurons in regulating female receptivity is not clear. Again, since 20E controls the metamorphic changes that occur in the CNS, it is not surprising that 20E would regulate the arborization of pC1 neurons. And since these neurons have been implicated in female receptivity, it would therefore be expected that altering 20E signaling in pC1 neurons would affect this phenotype. However, this does not mean that the defects in female receptivity expressed by PTTH mutants are due to defects in pC1 arborization. For this, the authors would at least have to show that PTTH mutants show the changes in pC1 arborization shown in Fig. 6. And even then the most that could be said is that the changes observed in these neurons "may contribute" to the observed behavioral changes. Indeed, the changes observed in female receptivity may be caused by PTTH/20E actions on different neurons.

C2. As newly formed prepupae, the ptth-Gal4>UAS-Grim flies display similar changes in gene expression to the genetic control flies to response to a high-titer ecdysone pulse. These include the repression of EcR (McBrayer et al., 2007). We tested whether there is a similar feedforward relationship between PTTH and EcR-A. We quantified the EcR-A mRNA level of PTTH -/- and PTTH -/+ in the whole body of newly formed prepupae. Indeed, PTTH -/- induced upregulated EcR-A in the whole body of newly formed prepupae compared with PTTH -/+ flies. We also detected the pattern of pC1 neurons when PTTH is deleted. Consistent with the feedforward relationship between PTTH and expression of EcR-A in newly formed prepupae, PTTH deletion induced less established pC1-d neurons contrary to that induced by EcR-A reduction in pC1 neurons.

However, it is not sure that the expression of EcR-A in pC1 neurons increases compared with genetic controls when PTTH is deleted. Furthermore, on the one hand, manipulation of PTTH has general effect on the neurodevelopment. On the other hand, the detailed pattern of pC1-b neurons which is the key subtype regulating female receptivity through EcR-A function in pC1 neurons could not be seen clearly. So, the abnormal development of pC1b neurons, if this is true, is just one of the possible reasons for the effect of PTTH deletion on female receptivity.

(3) Some of the results need commenting on, or refining, or revising: a- For some assays PTTH behaves sometimes like a recessive gene and at other times like a semidominant, and yet at others like a dominant gene. For instance, in Fig. 1D-G, PTTH[-]/+ flies behave like wildtype (D), express an intermediate phenotype (E-F), or behave like the mutant (G). This may all be correct but merits some comment.

C3. Female receptivity increases with the increase of age after eclosion, not only for wild type flies but also PTTH mutants. At the first day after eclosion (Figure 1D), maybe the loss of PTTH in PTTH[-]/+ flies is not enough for sexual precocity as in PTTH -/-. At the second day after eclosion and after (Figure 1E-G), the loss of PTTH in PTTH[-]/+ flies is sufficient to enhance female receptivity compared with wild type flies. However, After the 2nd day of adult, female receptivity of all genotype flies increases sharply. At the 3rd day of adult and after, female receptivity of PTTH -/- reaches the peak and the receptivity of PTTH[-]/+ reaches more nearly to PTTH -/- when flies get older.

b - Some of the conclusions are overstated. (i) Although Fig. 2E-G does show that silencing the PTTH neurons during the larval stages affects copulation rate (E) the strength of the conclusion is tempered by the behavior of one of the controls (tub-Gal80[ts]/+, UAS-Kir2.1/+) in panels F and G, where it behaves essentially the same as the experimental group (and quite differently from the PTTH-Gal4/+ control; blue line).(Incidentally, the corresponding copulation latency should also be shown for these data.). (ii) For Fig. 5I-K, the conclusion stated is that "Knock-down of EcR-A during pupal stage significantly decreased the copulation rate." Although strictly correct, the problem is that panel J is the only one for which the behavior of the control lacking the RNAi is not the same as that of the experimental group. Thus, it could just be that when the experiment was done at the pupal stage is the only situation when the controls were both different from the experimental. Again, the results shown in J are strictly speaking correct but the statement is too definitive given the behavior of one of the controls in panels I and K. Note also that panel F shows that the UAS-RNAi control causes a massive decrease in female fertility, yet no mention is made of this fact.

C4. (i) For all figures in the text, only when all the control groups were significant different from assay group, we say the assay group is significantly different. In Figure 2E-G, the control groups were both different from the assay group only at the larval stage. The difference between two control groups may due to the genetic background. We have described more detailed statistical analysis in the legend. In addition, the corresponding copulation latency has been shown. (ii) For Figure 5, we have revised the conclusion in text as “when the experiment was done at the pupal stage is the only situation when the controls were both different from the experimental.” Besides, the UAS-RNAi control causes a massive decrease in female fertility in panel F has been mentioned.

**Reviewer #3 (Recommendations For The Authors):**
(1) I am not sure that PTTH neurons should be referred to as "PG neurons". I am aware that this name has been used before but the PG is a gland that does not have neurons; it is not even innervated in all insects.

C5. Agree. “PG neurons” has been changed into “PTTH neurons”.

(2) Fig. 1A warrants some explanation. One can easily imagine what it shows but a description is warranted.

C6. Explanation has been added.

(3) When more than one genotype is compared it would be more useful to use letters to mark the genotypes that are not statistically different from each other rather than simply using asterisks. For instance, in the case of copulation latencies shown in Fig. 1E-G, which result does the comparison refer to? For example, since the comparisons are the result of ANOVAs, which comparison receives "*" in Fig. 1F? Is it PTTH[-]/+ vs PTTH[-]/PTTH[-] or vs. +/+?

C7. Referred genotypes and conditions were marked in all figure legends.

(4) Fig. 1H: Why is copulation latency of PTTH[-]/PTTH[-]+elav-GAL4 significantly different from that of PTTH[-]/PTTH[-]? This merits a comment. Also, why was elav-GAL4 used to effect the rescue and not the PTTH-GAL4 driver?

C8. We could not explain this phenomenon. This may due to the different genetic backgrounds between controls. We have mentioned this in figure legend.

(5) Fig. 2C, the genotype is written in a confusing order, GAL4+UAS should go together as should LexA+LexAop.

C9. We have revised for avoiding confusion.

(6) In Fig. 2, is "larval stage" the same period that is shown in Fig. 3A? Please clarify.

C10. We have clarified this in text and legends.

(7) Fig. 6. The fact that pC1 neurons can be labeled using the pC1-ss2-Gal4 at the start of the pupal stage does not mean that this is when these neurons appear (are born), only when they start expressing this GAL4. Other types of evidence would be needed to make a statement about the birthdate of these neurons.

C11. We have revised the description for the appearance of pC1-ss2-Gal4>GFP. The detailed birth time of pC1 neurons will be tested in future.

(8) The results shown in Fig. 7 are not pursued further and thus appear like a prelude to the next manuscript. Unless the authors have more to add regarding the role of one of the differentially expressed genes (e.g., dopamine beta-monooxygenase, which they single out) I would suggest leaving this result out.

C12. We have leave this out.

(9) Female flies lacking PTTH neurons were reported to show lower fecundity by McBrayer et al. (2007) and should be cited.

C13. This important study has been cited in the first manuscript. In this revision, we have cited it again when mentioning the lower fecundity of female flies lacking PTTH neurons.

(10) Line 230: when were PTTH neurons activated? Since they are dead by 10h post-eclosion it isn't clear if this experiment even makes sense.

C14. Yes, we did this for making sure that PTTH neurons do not affect female receptivity at adult stage again.

(11) Line 338: the statements in the figures say that PTTH function is required during the larval stages, not during metamorphosis

C15. This has been revised as “The result suggested that EcR-A in pC1 neurons plays a role in virgin female receptivity during metamorphosis. This is consistent with that PTTH regulates virgin female receptivity before the start of metamorphosis.”

(12) Did the authors notice any abnormal behavior in males? McBrayer et al. (2007) mention that males lacking PTTH neurons show male-male courtship. This may remit to the impact of 20E on other dsx[+] neurons.

C16. Yes, we have noticed that males lacking PTTH show male-male courtship. It is possible that PTTH deletion induces male-male courtship through the impact of 20E on other dsx+ or fru+ neurons. We have added the corresponding discussion.

(13) Line 145: please define CCT at first use

C17. CCT has been defined.

(14) Overall the manuscript is well written; however, it would still benefit from editing by a native English speaker. I have marked a few corrections that are needed, but I probably missed some.+ Line 77: "If female is not willing..." should say "If THE female is not willing..."+ Line 78 "...she may kick the legs, flick the wings," should say "...she may kick HER legs, flick HER wings,"+ Lines 93-94 this sentence is unclear: "...while the neurons in that fru P1 promoter or dsx is expressed regulate some aspects..."+ Line 108 "...similar as the function of hypothalamic-pituitary-gonadal (HPG).." should say "...similarTO the function of hypothalamic-pituitary-gonadal (HPG).."+ Line 152 "Due to that 20E functions through its receptor EcR.." should say ""BECAUSE 20E ACTS through its receptor EcR.."+ Lines 155, 354 "unnormal" is not commonly used (although it is an English word); "abnormal" is usually used instead.+ Line 273: "....we then asked that whether ecdysone regulates" delete "that" + Sentences lines 306-309 need to be revised.

C18. Thank you for your suggestions. We have revised as you advise.